# Case Study of Collaborative Modeling in an Indigenous Community

Gavin Wade McDonald [1,†], Lori Bradford [2,3,*,†], Myron Neapetung [4], Nathaniel D. Osgood [1], Graham Strickert [3], Cheryl L. Waldner [5], Kurt Belcher [3], Lianne McLeod [5] and Lalita Bharadwaj [6]

1  Department of Computer Science, University of Saskatchewan, Saskatoon, SK S7N 5C9, Canada
2  School of Professional Development, College of Engineering, University of Saskatchewan, Saskatoon, SK S7N 5A9, Canada
3  School of Environment and Sustainability, University of Saskatchewan, Saskatoon, SK S7N 5C8, Canada
4  Community Based Research Coordinator, Yellow Quill First Nation, Kelvington, SK S0A 3A0, Canada
5  Western College of Veterinary Medicine, University of Saskatchewan, Saskatoon, SK S7N 5B4, Canada
6  College of Medicine, University of Saskatchewan, Saskatoon, SK S7N 5E5, Canada
*  Correspondence: lori.bradford@usask.ca; Tel.: +1-306-966-1617
†  These authors contributed equally to this work.

**Abstract:** To support Indigenous communities in preparing for uncertainties such as climate change impacts and unexpected threats to health, there are calls by researchers and community members for decision support tools that meaningfully and sensitively bring together Indigenous contextualized factors such as social dynamics, local- and culture-specific knowledge, and data with academic tools and practices including predictive modeling. This project used a community engaged approach to co-create an agent-based model geographically bounded to a reserve community to examine three community-requested simulations. Community members and researchers co-designed, built, and verified the model simulations: a contaminated water delivery truck; a Pow Wow where a waterborne infectious disease spreads; and a flood which restricts typical movement around the reserve for daily tasks and health care. The simulations' findings, displayed as both conventional and narrative outputs, revealed management areas where community adaptation and mitigation are needed, including enhancing health service provision in times of disease outbreaks or large community events, and creating back-up plans for overcoming flood impacts to ensure services are accessible for vulnerable members of the community. Recommendations for communities, researchers, and modelers are discussed.

**Keywords:** agent-based modeling; participatory modeling; flood management; flood recovery; truck-to-cistern water systems; Indigenous communities; storytelling; water quality; epidemiology; health policy

## 1. Introduction

### 1.1. Land Acknowledgement and Positionality Statement

The authors of this paper live, and the work herein reported occurred, in Treaty 4 and 6 Territories in Saskatchewan, Canada, home of the Cree, Saulteaux, Nakota, Dene, Assiniboine and Ojibwa, and Métis peoples. The authors also acknowledge their privileges in education and socio-economic status and reflect that an Indigenous community-based research coordinator was involved in all steps and is a co-author for this work.

### 1.2. Context

Yellow Quill First Nation (YQFN) is a Nahkawininiwak (Saulteaux) community located in Treaty 4 territory, approximately 250 km east of Saskatoon, Saskatchewan, see Figure 1. The community is situated in a low-lying area adjacent to Pagāni-Sāgahigan (Nut Lake). YQFN is served by a reverse-osmosis water treatment plant; part of the community

has a piped water service whereas the rest have a truck-to-cistern service. The health effects of flooding and access to potable water are of great concern to the community due to their low elevation compared with surrounding regions in the Lake Winnipegosis watershed. The concerns raised with researchers were based on community-driven interest in adapting to continued experiences of spring flooding, as well as preparing for unpredicted flooding from extreme rain events expected to occur more often due to climate change [1,2]. There is a need for researchers to collaborate with Indigenous communities, such as YQFN, regarding preparations for climate change effects and potential disease outbreaks, and to support decision processes with co-developed and culturally informed tools [3–8]. Additionally, chronic underfunding of health services on reserves has led to situations where communities may be at heightened risk of impacts to health and wellbeing simply because of being short-staffed [9,10].

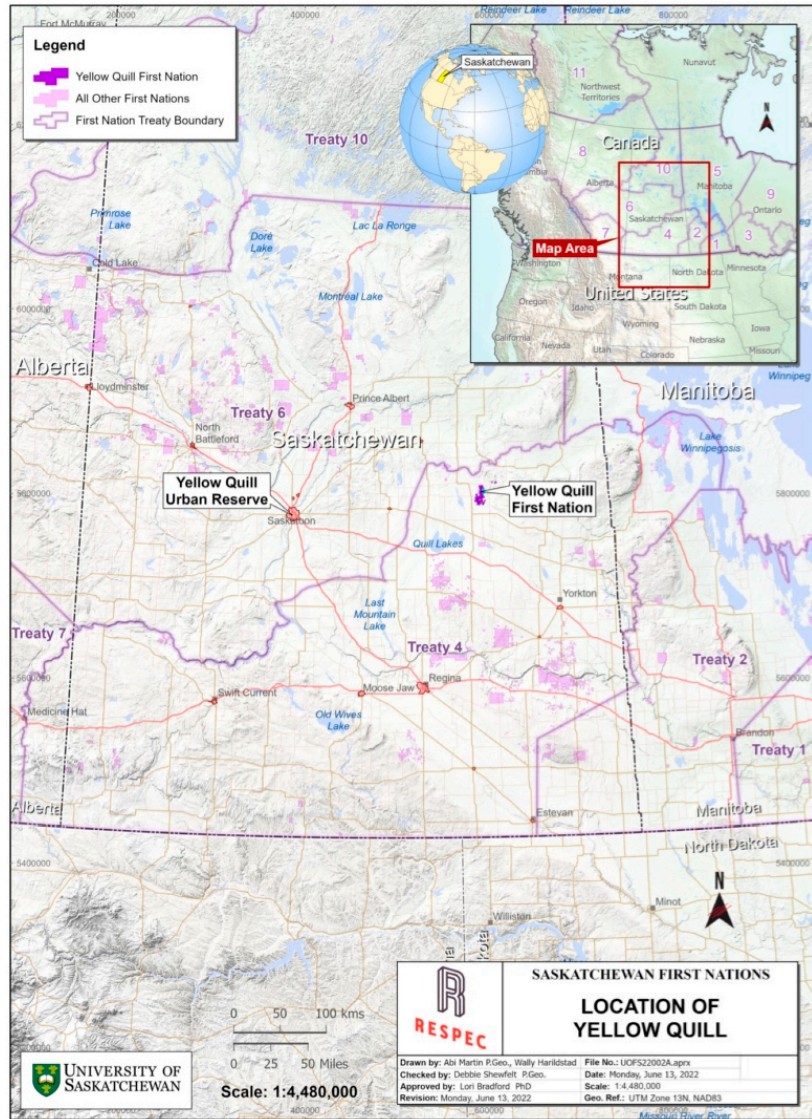

**Figure 1.** Location of Yellow Quill First Nation in Treaty 4 territory.

## 1.3. Agent-Based Modeling

Researchers in a variety of natural resource and health fields are advocating for participatory modeling as a response to challenges encountered with conventional decision support systems and stakeholder engagement methods for predictive analytics and preparing for complex problems [11–13]. Agent-based modeling has been put forward to connect stakeholder engagement and integration of predictive data and computational models in

specific contexts [14,15]. Agent-based modeling is a dynamic simulation modeling method where complex systems are characterized as an environment occupied by one or more populations of interacting, autonomous actors termed agents [16]. The method involves placing a collection of one or more types of agents into a simulated environment, running the simulation over time in various scenarios, and observing emergent behavior of the system. Agents can be characterized as being unique, in that they have one or more characteristics that might differ from agent-to-agent, situated at a logical, physical, or social position in the simulated environment, and autonomous, in that they act independently from one another [16,17]. Each agent may retain a state that can include memory of its own history and may adapt its behavior in response to its own state and characteristics, interactions with other agents, and its environment [16,17].

Agent-based models (ABMs) can be valuable tools for facilitating investigation of the potential impacts of health and policy interventions within complex systems. Compared with other simulation methodologies, ABMs can readily incorporate multiple dimensions of heterogeneity among people and other elements within a system [18,19]. ABMs are intrinsically stochastic, in contrast with some other methods that are deterministic. They can further readily represent stylized to highly textured environments, including geography. These models have been used in Indigenous contexts (i.e., for modeling disease outbreaks, hunting, subsistence agriculture) to support decision making for public health or ecology [20,21].

The intuitive and graphical nature of agent-based models also make them valuable for participatory modeling approaches. Involvement of stakeholders in model conceptualization and parameterization can improve models by including stakeholder expertise and context [22]. Furthermore, community stakeholders and decision makers are often well positioned to inform the development of practical and feasible scenarios to be modelled [23]. In turn, such models are valuable tools for community engagement and decision making. In this context, models do not have to precisely represent reality or predict future outcomes; rather, they provide a means to explore scenarios and potential interventions as a means of developing a collective understanding and potential actions to address a particular health or other community challenge [22,23]. In this project, stylized agent-based models were developed in collaboration with the YQFN to address community concerns about water quality and flooding and health service needs.

## 1.4. Discrete Event Simulation

Discrete event simulation (DES) is a simulation methodology concerned with queueing and resource-constrained workflows. The building blocks of DES are entities, events, resources, and queues. As entities move through the model, they use resources; if a needed resource is in use, the entity must form a queue and wait for it to become available. Entities may interact with each other and experience events which change their attributes, or state. Like ABM, evaluation of a DES model is focused on the observation of emergent behavior from the system [24].

## 1.5. System Dynamics Modeling

System dynamics (SD) is a modeling method where models are built from stocks, flows, and auxiliary variables. The state of the system at any given time can be described by the stocks. The rate of change of the system state is determined by the flows. Feedback loops are a key unit of analysis [25]. SD models have a one-to-one correspondence with aggregate models constructed from ordinary differential equations (ODEs).

Compared with SD, ABM and DES make it simpler to implement heterogeneity in the model but compute performance scales more poorly with the population. Some properties of different simulation methods are summarized in Table 1.

**Table 1.** Simulation methods. Adapted with permission from [25], © Elsevier, 2022.

|  | SD | DES | ABM |
|---|---|---|---|
| Perspective | Top-down | Top-down | Bottom-up |
| Stochasticity | Deterministic | Stochastic | Stochastic |
| Building Blocks | Stocks, flows, feedback | Entities, events, queues | Agents, decision rules |
| Development time | Low | High | High |
| Population Scalability | Good | Poor | Poor |
| Heterogeneity | Complex | Simple | Simple |
| Network Effects | No | Yes | Yes |

*1.6. Objectives*

This project was co-conceived by community representatives and researchers to inform local decision-making processes through three objectives:

1. Grounded by diverse data sources, develop a model framework with ABM to assess and investigate comprehensive impacts on the community members from flooding,
2. Demonstrate the capability of ABMs as an operational tool for evaluating and supporting health services and emergency planning and management measures, and,
3. Contribute to the sustainability of the community and their environment by providing a tool to investigate complex interactions and feedbacks between human and natural systems and to communicate understanding of flooding impacts and improvements to mitigation measures.

## 2. Materials and Methods

*2.1. Ethics*

This research was conducted under the approval of the University of Saskatchewan Behavioural Research Ethics Board ethics number 18-08, for Principal Investigator Lalita Bharadwaj. The research was also approved by the Chief and Council members of Yellow Quill First Nation, and a community feast and a blessing by an Elder occurred prior to any research activities occurring.

*2.2. Community Engagement*

Members of the research team (Bharadwaj, Bradford) have been engaged for over ten years with Yellow Quill First Nation leadership and community members on community-driven projects regarding water, from securing access, recognizing values of local waterbodies, and monitoring quality and quantity of local surface and groundwater, to understanding threats to local waters used for drinking, harvesting fish, cultural activities, maintaining hydrological flows, and valuing the ecosystem services provided by water [26–28]. A co-designed research plan, as well as ongoing strong individual relationships with community leaders meant that the project emerged respectfully, and local data was freely shared through established ownership, control, access, and protection agreements. During one of the research results meetings for previous projects, community members asked specifically to examine what would happen to infrastructure and community dynamics with various levels of flooding, and who would be most vulnerable. Community members and councilors were interested in ensuring that emergency needs, such as evacuation routes and flood-safe storage sites for equipment, were identified. They were also interested in exploring how a contamination event in the truck-to-cistern supply system may spread illness. Because of the nature of these questions, and the community's familiarity with modeling from previous projects, an agent-based model was suggested as a way to develop a foundational model from which to explore multiple scenarios.

Once the funding decision was received, regular research progress meetings occurred bi-monthly over two years with the Chief and Council members of the community. A locally based community research coordinator (Neapetung) was hired to liaise with the researchers (Bharadwaj, Bradford, Belcher) and coordinate data gathering events. Eight data gathering events occurred between March 2017 and August 2019 in the community. These events each started with community feasts at the local community hall as requested as part of the

community engagement strategy, followed by individual and group-based stations set up for data gathering (community asset mapping, interviews, sharing circles, and arts-based methods). Maps of the community were provided and members drew their regular daily activities at different locations, and the approximate timing of their activities. Technicians from the community who worked on operations, roads, lands, and resources portfolios were also made available to consult with a computer science student, (MacDonald) who was building the agent-based model, about technical questions such as water system design, infrastructure weaknesses, and climate. The later community gatherings included a researcher (Bradford) setting up a laptop and screen with initial versions of the model. Community members were asked to analyze the model. When suggested by community members, more data was collected (i.e., on the school bus route, and on the frequency of visits to Elders' homes). Once data were aggregated, a community meeting was arranged (August 2019) for verification of overall themes, specific sites of concern, and community dynamics (i.e., where most people travelled and interacted throughout a 'typical day'). At the verification meeting, community members noted that charts and tables were too scientific, thus, the suggestion to add a narrative diary of daily activities for each agent was incorporated into the model outputs.

### 2.3. Simulation Model

The agent-based simulation model (ABM) was constructed using AnyLogic software [29], a multi-method simulation modeling package employing the Java [30] programming language. The model includes a geographic information systems (GIS) component that allows agents to appear on a map of the YQFN community. The model includes representations, as agents, of people, homes, workplaces, schools, cultural sites, nursing stations, water treatment and distribution infrastructure, and areas ponded due to flooding. An embedded discrete event simulation model simulates patient flow at a nursing station in the community; this arose due to a request from community members to understand what additional health resources would be needed if a waterborne illness occurred. A generalized health condition, which can represent communicable or chronic disease, was also implemented so that the community could observe potential rates of community-spread or impacts on access to care. The model also incorporated a storytelling feature that permits following key events in a particular agent's experience during a simulation. The storytelling feature was requested as it best aligned with the oral storytelling tradition as a key means of education in the Saulteaux culture. This model can be considered 'reactive' as opposed to 'thinking', as no sophisticated cognitive processes are modeled in the person agents.

### 2.3.1. Model Purpose and Scope

The model simulates the potential effects of various scenarios, designed to facilitate discussion and reasoning related to water security and health in engagement with the community. This work does not propose to provide predictive modeling for specific emergencies or real-time decision analytics, rather it provides a coarse-grained comparison of the extent of outcomes from certain plausible scenarios. Whereas some works [31,32] examine cooperative or adversarial interactions between multiple stakeholders, this work focuses on a single stakeholder group, the community of YQFN. Because water is delivered to some household cisterns by truck, the first scenario simulates the effects of microbial contamination of a water delivery truck on the spread of waterborne pathogens throughout the community. This scenario is also informed by research in other communities [8,33]. A second scenario simulates the spread of waterborne pathogens throughout the community if contaminated water were to be served at a Pow Wow (a major community event). Each of these scenarios tracks the number of people exposed to the pathogen, infectious with the pathogen, and seeking health care due to illness. The third scenario simulates the impact of flooding on preventing access to health care in the community due to ponding and road washouts commonly experienced in the community during heavy rainfall events.

This scenario monitors the proportion of people experiencing impaired mobility within the community, and therefore impaired access to health care.

2.3.2. Agents

Person agents are distinguished by age, sex, geographic position, home and work location, and care-seeking probability. The population of person agents was designed to align with current community profiles on population demographics, which were available through federal census, local census reporting, and what researchers heard at local data gathering events about the make-up of the community. Economic, cultural, and ethnic differences were not considered but remain open as avenues of future work given interest and data availability. A person may be considered healthy, with no health conditions, or may have one or more generic health conditions that may cause them to seek health care. Each person will normally commute from home to work or school and back each day; this may be interrupted by seeking care for a health condition or attending a cultural event in the community.

Homes, workplaces, a school, a nursing station (clinic), water treatment infrastructure, and cultural sites are represented in the model and assigned a geographic location on the community map. The model tracks person agents present at each place at any given time, and each place is associated with a water source. Total water use for each place is computed based on the number of people inside and the water use per person per day and influences how often the water truck visits the place. A place may be clean or contaminated with pathogens and may pose other generalized health hazards such as injury to the occupants. Each health hazard, including contamination by waterborne pathogens, carries with it a specified probability per six-hour period that a person in that place will be affected by the associated health condition.

Patient flow at the nursing station is represented by a discrete event simulation model [24], in which patients arrive and wait for treatment. Patients may become tired of waiting and leave before being seen or stay and eventually be allocated a health worker and bed and then undergo treatment.

Water sources are modeled as agents and can represent treatment plants, reservoirs, or cisterns. The water treatment plant is associated with a treatment capacity, which influences the rate at which its reservoir refills, and may be contaminated with a waterborne pathogen. Cisterns serve a single building and enter a queue for water delivery when their water levels drop below a certain level.

The water delivery truck agent represents a water truck that delivers water to houses on a truck-to-cistern system that forms part of the YQFN water distribution system. The water truck sits idle when no cisterns are empty, and no customers are calling for water. Trucks will move to make deliveries in the order the requests are received once triggered by low cistern water levels. The water truck has a specified capacity and will return to the filling station as required. If the truck's filling station runs out of water, it will wait for more water to be available. For particular scenarios, the water truck will be contaminated with a waterborne pathogen, which it will then spread to the cisterns that it serves with 100% probability when it fills the cistern. The truck could also undergo decontamination at a certain frequency; if a contaminated truck underwent such decontamination, it was assumed that the contamination would be eliminated. Questions around water contamination represent pervasive issues on First Nations in Canada [33,34] and were of great interest to the YQFN community.

A storyteller agent was implemented to aggregate key events happening to a single person agent during a model run. This agent again lacks true agency but creating it as an agent allows for convenient implementation using the AnyLogic graphical editor. This feature was implemented at the request of community members to present results in a way that aligned with the significance of storytelling in First Nations' culture.

The population of storyteller agents exists at the root level of the model. Each is associated with a person either randomly or as chosen by the user. By default, 2% of person agents have a storyteller. When a person agent experiences a key event, it invokes the

tellMyStory method which records the event in the associated storyteller, if it exists. If no storyteller agent is associated with that person agent, then nothing is recorded.

Figure 2 shows the storyteller in the running model. The person agent whose story is being told began by following their regular commute between home and work; this person was exposed to an infectious agent on day 5 but was not infected, and again on day 7 where they contracted the waterborne illness. On day 8, the person attended the nursing station/clinic for treatment.

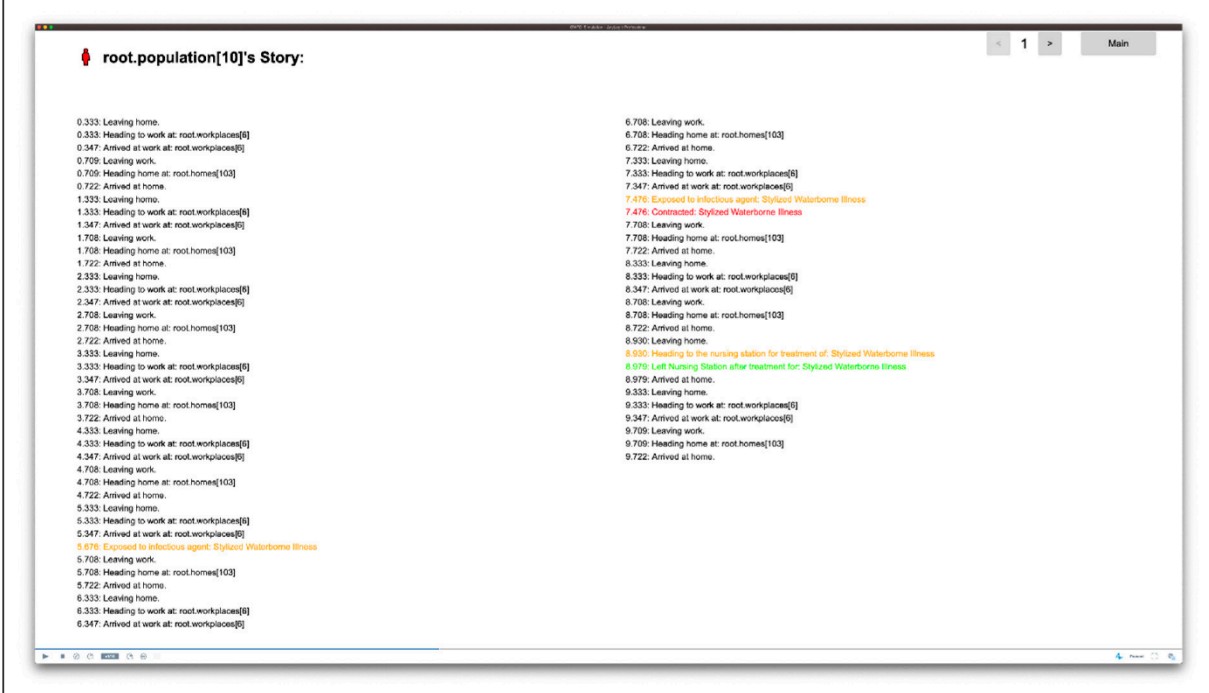

**Figure 2.** A screenshot of the storyteller agent documents the daily events and movements for a person agent over the course of a model run. Colored entries represent the events involving exposure to an illness or seeking treatment (orange), contracting the illness and experiencing symptoms (red), and recovering from the illness with or without professional treatment (green).

### 2.3.3. GIS Environment

Agents in the model exist in a geographic (GIS) environment, appearing and moving around on a geographic map of YQFN, verified by community members. People and other mobile agents follow actual scale roads, where available, and move between origins and destinations within the community as per the daily routines described and collected during community engagement events. Buildings, water treatment infrastructure, and cultural sites are placed at their actual map locations as determined from an orthographic air photo. AnyLogic provides for loading of the base map from OpenStreetMap [35], as well as placing overlays in a shapefile format. A shapefile is overlaid to indicate the extent of the YQFN lands.

The use of precise GIS in a stylized model provides several benefits. One is that it allows stakeholders to see the simulation playing out over a map of their own community. Another is that agent movements can be pathed along roadways. Finally, it allows the flooding data, which is georeferenced, to be integrated, as discussed in Section 2.3.4.

### 2.3.4. Flooding

Flooding in the community is represented by areas of ponding associated with heavy precipitation. Ponding data were obtained from the output of the wetland DEM ponding model (WDPM) developed by the University of Saskatchewan Centre for Hydrology, where DEM stands for digital elevation model, and based on community-requested values of 10-, 50-, and 100-year flood events [36,37]. Within the ABM, each ponded area is associated with

an agent that periodically checks to see if any person agents are within its bounds; a message is sent to each person agent who has entered the ponded area, and, if appropriate for the scenario, the person will stop their movement and indicate that they are blocked. Community members verified sites where driveways and roads become impassable during heavier floods.

### 2.3.5. Health Conditions

The model represents health conditions as agents governed by generalized state charts that can be parameterized to represent either communicable or non-communicable diseases. Communicable diseases can be set to progress through pre-infectious (latently infected), infectious, and post-infectious states, whereas non-communicable diseases move directly to the post-infectious state. Chronic diseases stay in the post-infectious state indefinitely. For the purposes of the scenarios investigated in this study, three stylized heath conditions were implemented: a waterborne illness that is transmissible through contaminated water and person-to-person, a generic physical injury, and type II diabetes requiring dialysis. Type II diabetes was chosen as one of the health conditions due to its disproportionate impact on Indigenous peoples in Canada as compared with the general population [38] and community interest. Further, Indigenous people with youth-onset diabetes are at greater risk for end-stage renal disease and death than the general population [39]. The general physical injury represents another example of the configuration of the health condition as well as an additional motivation for care-seeking behavior by person agents. An additional state chart represents treatment status, which may reduce severity of symptoms.

The representation of waterborne illness follows the susceptible exposed infective recovered susceptible (SEIRS) model [39] with the baseline scenario characterized by a latent period of one day, an infectious period of 3 days, waning of immunity after 14 days and a rate of person-to-person transmission of 25% per 0.1 days. An infected person will remain in their home, except for seeking health care, for the balance of the 5-day duration of their illness. This is not currently calibrated to match a specific real-world pathogen and instead represents a stylized illness and allows reasoning about behavior in diverse scenarios involving the community and consultation and collaboration with community stakeholders.

Type II diabetics in the model require dialysis about three times per week. At the conclusion of this interval, their type II diabetes health condition will enter an untreated state and they will attempt to seek care at the clinic.

A person with a health condition in the untreated state will, at a rate of 1.0 per day, attempt to seek care with a probability given by that person's care seeking probability parameter. At a population level, the care seeking probability is uniformly distributed from 0.0 to 1.0.

The model structure is described in greater detail in Appendix A.

### 2.4. Scenarios and Parameterization

### 2.4.1. Scenarios

Broadly, three scenarios were postulated in the current version of the model: a contaminated truck, Pow Wow, and impacts of flooding on mobility and unmet care needs of people within the community. Although the model was designed with a broad framework to allow exploration of diverse scenarios, the cross-cutting concern joining these scenarios is community access to health services.

The contaminated truck scenario examines the spread of disease if the water delivery truck was contaminated with a pathogen causing waterborne illness, and the resultant impact on care seeking at the nursing station. As people get sick, they will attempt to seek care at the clinic, which was modeled at levels reflective of actual staffing. Wait times at the clinic depend on the number of patients and resources available. Patients who are treated will self-isolate until recovered, but those who leave without being seen will continue about their daily routine, potentially transmitting infection. This scenario runs for 28 days.

The Pow Wow scenario considers similar impacts to the contaminated truck scenario if contaminated water were to be served at a large community event. This scenario also runs for 28 days.

The mobility scenario simulates how a flooding event may interfere with the movement of people about the community, particularly the number of people prevented from seeking care for type II diabetes. This scenario runs for 5 days. This scenario emerged from initial sharing circles with Elders where they reported fear for losing their mobility to get to medical appointments, get children to school, and for community members to get to employment due to roads being washed out.

Time is continuous with units of days for all scenarios.

### 2.4.2. Parameters

Model parameter values were developed in consultation with the community, with some population parameters—including the age distribution—being derived from 2016 Canada Census data [40].

The water truck scenario assumed that a single water truck served the community and would become contaminated with a pathogen and subsequently contaminate any cisterns it filled. In the Pow Wow scenario, it was assumed the entire community would attend. In both the water truck and gathering scenarios, it was assumed that the hypothetical pathogen would impose a 75% probability of infecting each exposed person per 6 h. It was also assumed that all people requiring care would seek it, including those who became ill with the waterborne pathogen, those with injuries, and diabetics requiring dialysis. Diabetics would require dialysis approximately three times per week. The nursing station was assumed to be staffed with one health care worker, and the time until someone seeking care would leave without being seen was drawn from a continuous uniform distribution with a minimum of 1 and a maximum of 6 h. Once health care is received, the agent will stay home for the duration of their illness. The baseline values for parameters important to the outcomes in each of the described scenarios but which were not varied in the experiments are summarized in Table 2. Additional details about model parameterization and structure are given in Appendix A.

**Table 2.** Baseline model parameters not varied across experiments.

| Parameter | Baseline Value |
|---|---|
| **Demographics** | |
| Population | 800 [1] |
| Male: Female ratio | 1.0 |
| Unemployed fraction—Male | 0.5 [2] |
| Unemployed fraction—Female | 0.15 [2] |
| Prevalence of diabetes requiring dialysis | 0.02 |
| **Water Delivery** | |
| Probability contaminated truck results in contaminated cistern | 1.0 |
| **Waterborne Illness Natural History** | |
| Rate of infection by waterborne pathogen from contaminated premises | 0.75/6 h |
| Latent period | 1 day |
| Infectious period | 3 days |
| Rate of person-to-person transmission | 0.25 per 0.1 days |
| Duration of immunity | 14 days |
| **Clinic Operations** | |
| Rate ill person will seek care | 1.0/day |
| Time until leaving clinic without care | 1.0–6.0 h |
| Number of health care workers | 1 |
| Number of beds available | 4 |
| Time for appointment | 5 min |
| Time using bed after appointment | 0 min |

[1] Population drawn from probability distribution based on population pyramid from 2016 Canada Census of Population [10]. [2] Employment rates derived from community consultation.

Parameters varied for experiments included whether the illness could be transmitted person-to-person, the duration of illness, how often the water delivery truck was

cleaned, and rainfall amount. The baseline (default) values for each of these parameters are summarized in Table 3.

**Table 3.** Baseline values of parameters that were varied in experiments.

| Parameter | Baseline Value |
|---|---|
| Rate of person-to-person transmission | 0.25/0.1 days |
| Duration of waterborne illness | 5 days |
| Probability delivery truck is contaminated | 0.0 |
| Truck decontamination | Never |
| Rainfall | 0 mm |

*2.5. Outcome Measures*

For the contaminated water truck and Pow Wow scenarios, the proportions of persons in various states of exposure to and illness with respect to the waterborne pathogen were reported, specifically the proportions of susceptible, exposed, infected, and recovered persons. The model also outputs counts of persons in each health status (healthy, waterborne illness, diabetes, or injury) and cumulative counts of people treated at the nursing station for each health status. Cumulative counts of people treated (for any condition) and people who leave without being seen (LWBS) at the nursing station are also displayed.

For the flooding and mobility scenario, the model displays the fraction of people in the general community whose movement is blocked by flooding and the count of people unable to access transport to seek dialysis for type II diabetes. In fact, dialysis is not available at the local clinic; patients must be transported to another community to receive treatment. For the model, it was assumed that, if a flooding event prevented a patient from reaching the local clinic, they would also be unable to access transport.

*2.6. Experiments*

Table 4 lists the variations in each scenario that were run as experiments. This can be viewed as four experiments: one examining the contaminated truck effects with different disease characteristics; one examining the contaminated truck with different decontamination schedules; one looking at the Pow Wow with different disease characteristics; and one considering movement with different flooding levels. Similar to scenario development, the contaminated truck and Pow Wow experiments were not calibrated to a known pathogen, but represent stylized variations meant to explore outcomes with different pathogens and management practices. For each configuration of the experiments listed in Table 4, a Monte Carlo ensemble of 100 realizations of the model was run.

**Table 4.** Summary of the scenario variations from baseline comprising each of the experiments.

| Scenario | Variation | Description |
|---|---|---|
| Contaminated Truck | Baseline [1] | Baseline scenario |
| | Alt 1 | Disease is not transmissible person-to-person |
| | Alt 2 | 2-day longer illness duration |
| | Alt 3 | Disease is not transmissible person-to-person; 2-day longer illness duration |
| | Cleaning 1 | Truck is decontaminated daily |
| | Cleaning 1 Alt 1 | Truck is decontaminated daily; disease is not transmissible person-to-person |
| | Cleaning 5 | Truck is decontaminated every 5 days |
| | Cleaning 5 Alt 1 | Truck is decontaminated every 5 days; disease is not transmissible person-to-person |
| Pow Wow | Baseline [1] | Baseline scenario |
| | Alt 1 | Disease is not transmissible person-to-person |
| | Alt 2 | 2-day longer illness duration |
| | Alt 3 | Disease is not transmissible person-to-person; 2-day longer illness duration |
| Movement | Baseline [1] | No flooding |
| | 10 mm | Flood due to 10 mm precipitation event |
| | 20 mm | Flood due to 20 mm precipitation event |
| | 100 mm | Flood due to 100 mm precipitation event |

[1] See Table 3.

## 3. Results

### 3.1. Contaminated Truck Scenario

#### 3.1.1. Model Outcomes

The contaminated truck scenario simulates a stylized waterborne illness, as described in Section 2.3.4, being spread by the water delivery truck to cisterns from which people consume water and get sick. It is important to emphasize again that this is a stylized disease, and it has not been calibrated to match its behavior to a specific pathogen. Figure 3 shows model outputs from a single model realization using the baseline parameters for this scenario. The simulated disease spreads through most of the community over a period of about 2 weeks, Figure 3a. Figure 3b illustrates patient access to the nursing station; since the disease spreads gradually through the community, the nursing station can handle all visits and no patients leave without being seen. Whereas Figure 3 depicts the outcomes of a single realization, reflective of the stochastics in the model, the next section considers outcomes from ensembles of realizations.

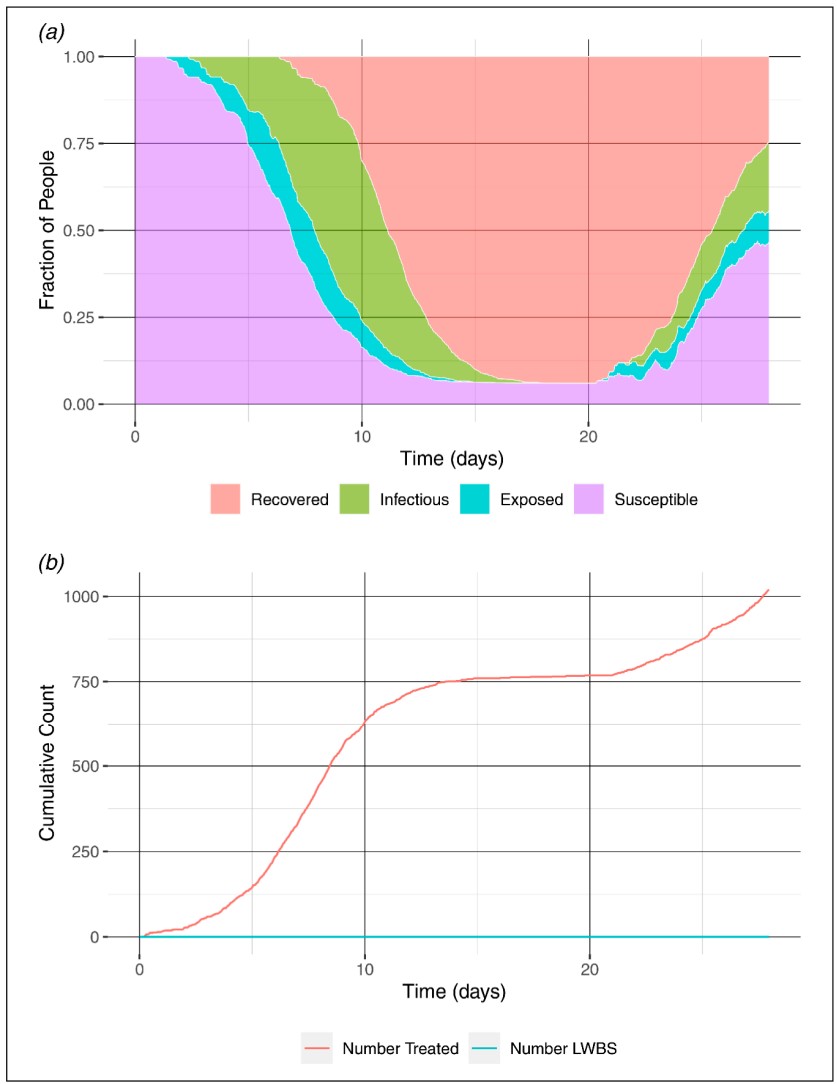

**Figure 3.** Results from contaminated truck scenario: (**a**) fraction of the population in susceptible, exposed, infectious, and recovered states; (**b**) cumulative count of people treated and leaving without being seen (LWBS) at the nursing station.

### 3.1.2. Ensemble Results

Figure 4 shows outputs from the contaminated truck scenario varying disease characteristics. The line represents the mean and the bands the 5th and 95th percentiles over 100 realizations of the model. Panel (a) depicts patients treated with baseline disease characteristics. In the experiments without person-to-person transmission, (b) and (d), fewer people were infected and sought treatment because the only way to acquire the disease was to be present at a place with a contaminated cistern, which were mostly residences. A longer disease duration resulted in a flatter curve in the case where there is person-to-person transmission (c) but had little effect when there was no person-to-person transmission (d).

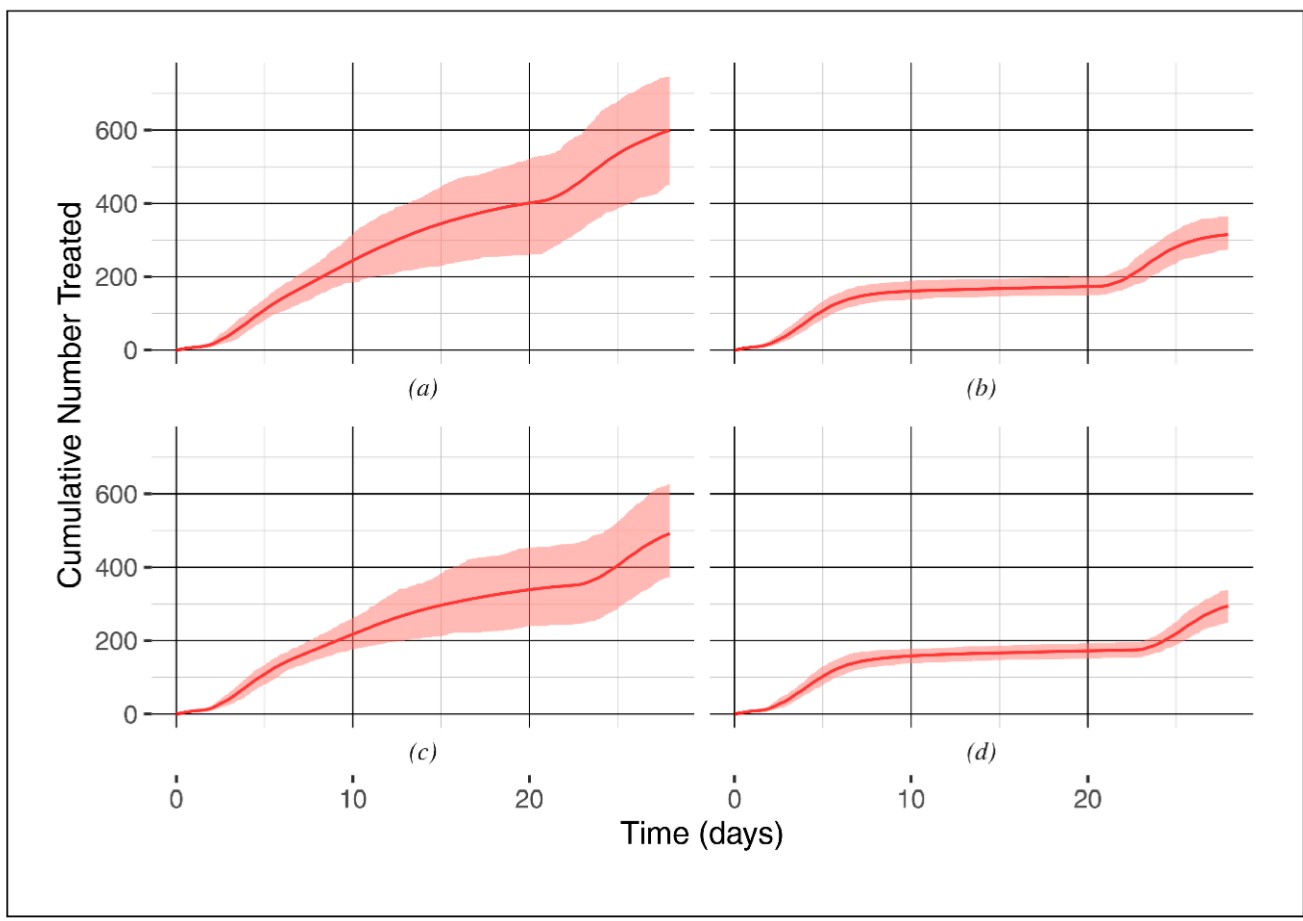

**Figure 4.** Patients treated from contaminated truck scenario: (**a**) baseline; (**b**) no person-to-person transmission; (**c**) longer duration of infection; (**d**) no person-to-person transmission and longer duration of infection. Line represents the mean and bands represent the 5th to 95th percentile over 100 realizations of the model.

Figure 5 shows patients treated from the contaminated truck scenario with varying decontamination interval and disease properties. Daily decontamination resulted in a noticeable reduction of infections for both the scenario with person-to-person transmission (a) and without (b). The 5-day decontamination interval was less effective in both the case with person-to-person transmission (c) and without (d). Pathogens may persist in the contaminated cisterns [33,41–43].

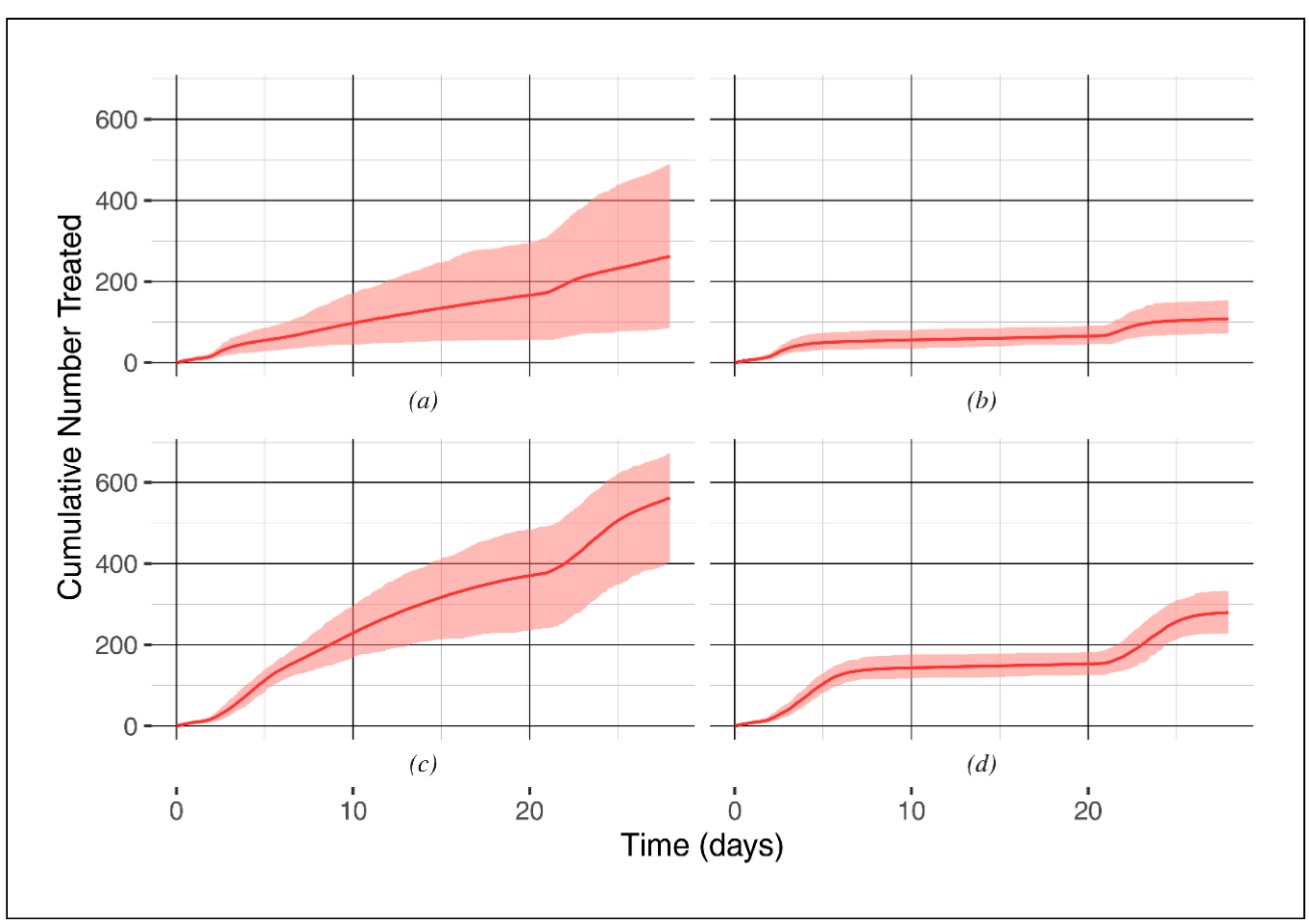

**Figure 5.** Patients treated from contaminated truck scenario: (**a**) 1-day cleaning interval with person-to-person transmission; (**b**) 1-day cleaning interval, no person-to-person transmission; (**c**) 5-day cleaning interval with person-to-person transmission; (**d**) 5-day cleaning interval, no person-to-person transmission. Bands represent the 5th to 95th percentile over 100 realizations.

### 3.2. Pow Wow Scenario

### 3.2.1. Model Outcomes

The Pow Wow scenario simulates a one-day cultural event where the entire community attends, and drinking water is contaminated. Figure 6 shows model outputs from one run of the Pow Wow scenario using baseline parameters. Figure 6a illustrates that most of the community became ill within a few days of the event. Since many people get sick at the same time, the nursing station is overwhelmed, as indicated by the substantial number of people leaving without being seen (Figure 6b).

### 3.2.2. Ensemble Results

Figure 7 shows outputs from the Pow Wow scenario when varying disease characteristics. Without person-to-person transmission, (b), fewer people were infected and sought treatment, but the outbreak still constituted a major disease event because the entire community attends the Pow Wow. Longer infection duration increased the number of patients treated where there was person-to-person transmission, (c), but had negligible effect where there was not, (d).

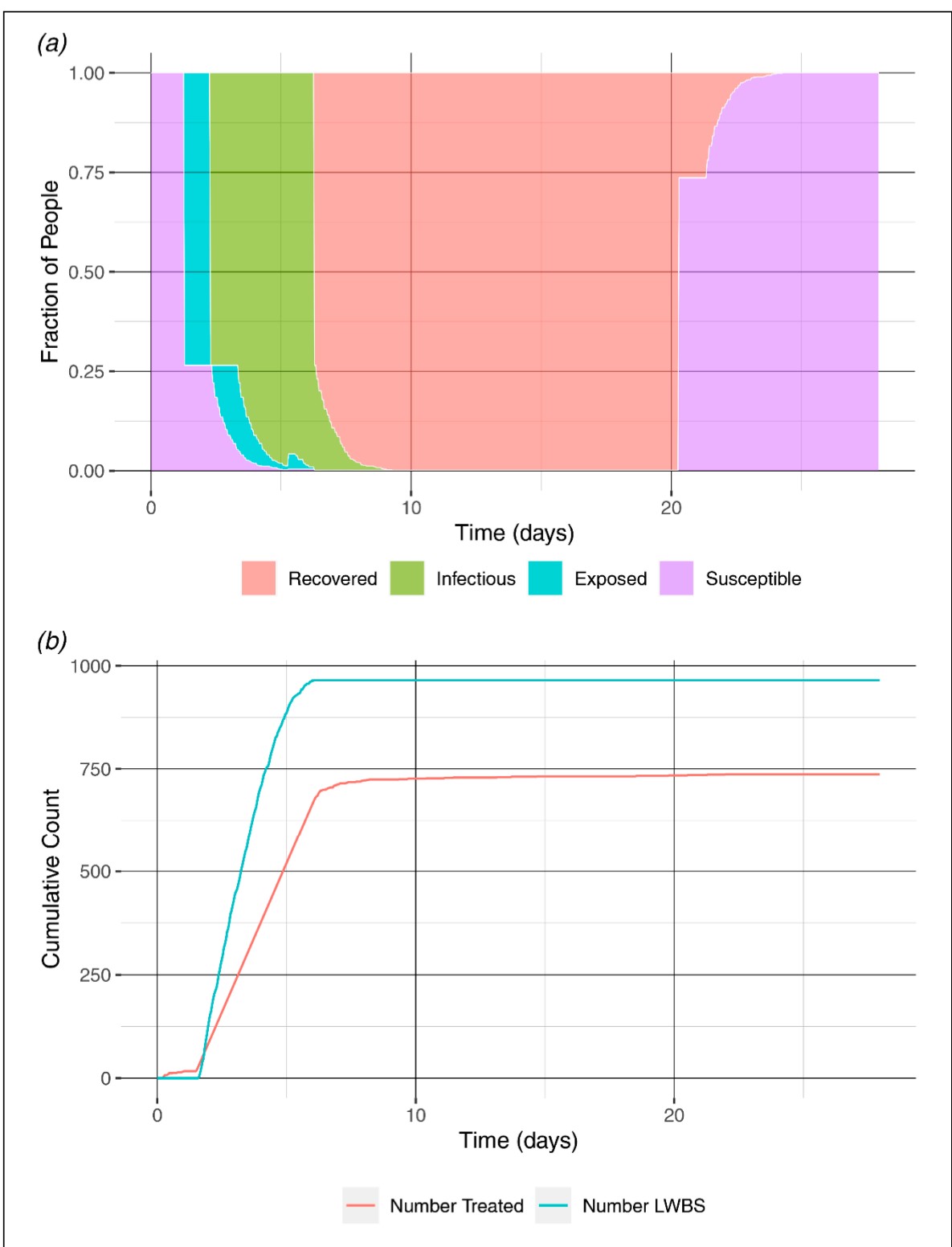

**Figure 6.** Results from Pow Wow scenario: (**a**) fraction of the population in susceptible, exposed, infectious, and recovered states; (**b**) cumulative count of people treated (blue) and leaving without being seen at the nursing station (red).

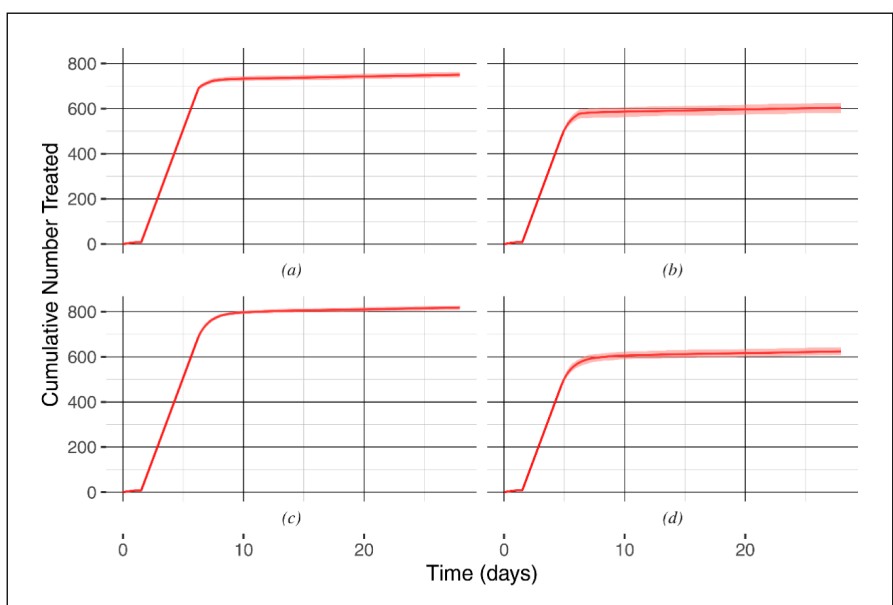

**Figure 7.** Patients treated from Pow Wow scenario: (**a**) baseline; (**b**) no person-to-person transmission; (**c**) longer duration of infection; (**d**) no person-to-person transmission and longer duration of infection. Bands represent the 5th to 95th percentile over 100 realizations of the model.

### 3.3. Mobility Impacts from Flooding Scenario

### 3.3.1. Model Outcomes

The mobility scenario simulates a flooding event and examines how it interferes with the movement of people about the community. Figure 8 shows the model map view for the main area of the community for the mobility scenario with a 100 mm precipitation event flood levels; blue polygons indicate areas with ponded water due to flooding, as obtained from the WDPM [36,37]; persons represented in red have been blocked in their movement. In this example, about 45% of the community residents have had their movement interrupted by the flood.

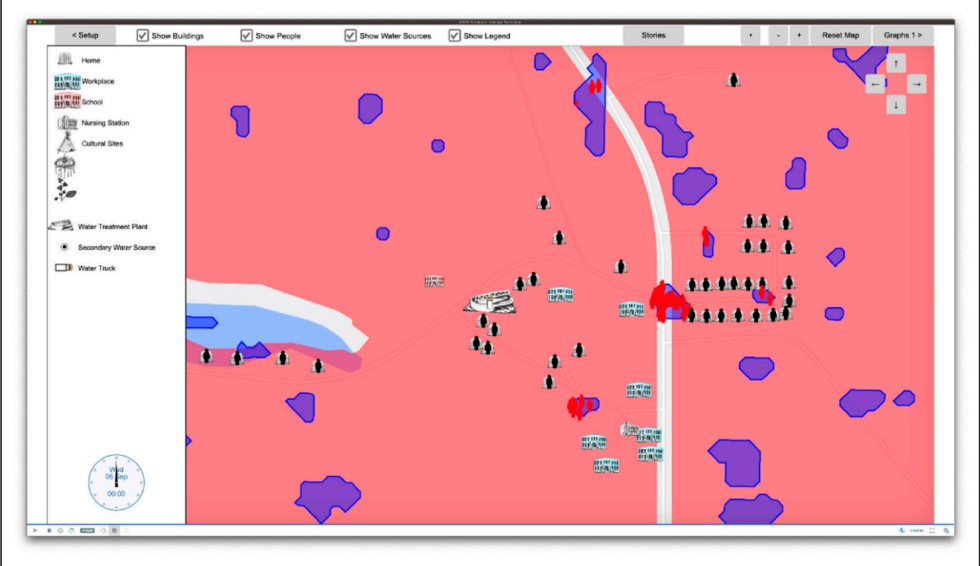

**Figure 8.** Mobility scenario for 100 mm precipitation event. Red color represents the land within the community boundaries. Blue color represents ponded areas due to flooding. Red agents represent those whose movement is blocked by the flooding event.

### 3.3.2. Ensemble Results

Figure 9 shows the mean number of people whose movement was blocked among dialysis patients and Figure 10 shows the number of people whose movement was blocked among the general population over the 5-day simulation time horizon. Heavier rainfall events led to more flooding and disruption of movement within the community.

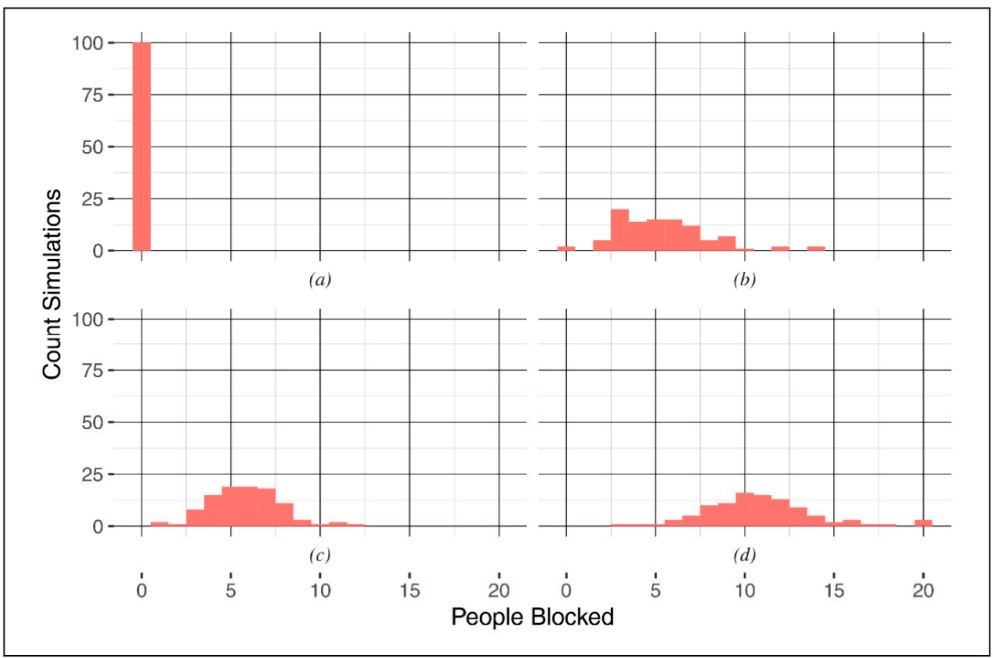

**Figure 9.** Dialysis patients untreated and blocked: (**a**) baseline; (**b**) 10 mm precipitation flood event; (**c**) 20 mm precipitation flood event; (**d**) 100 mm precipitation flood event.

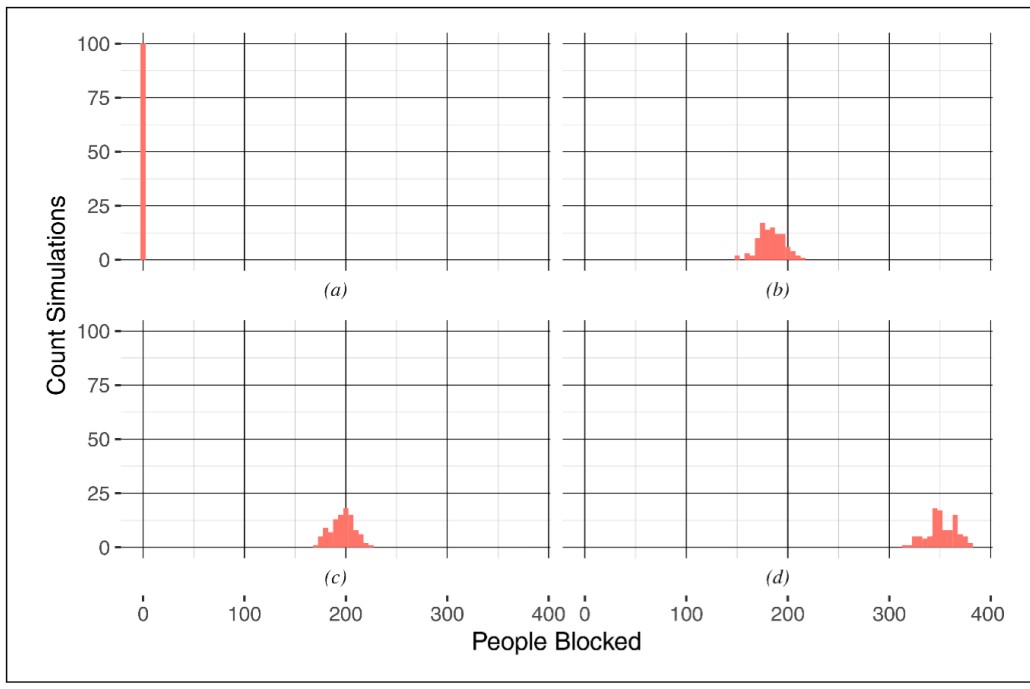

**Figure 10.** General population blocked: (**a**) baseline; (**b**) 10 mm precipitation flood event; (**c**) 20 mm precipitation flood event; (**d**) 100 mm precipitation flood event.

## 4. Discussion

The objectives of this work included co-creating a decision support tool that was grounded by diverse data sources (academic and community-based, experiential, and predictive, quantitative, and qualitative) and that could evaluate potential adaptations to climate change impacts such as increased flooding and health threats, and service needs during emergencies. Given the community's request to consider connections between human and natural systems holistically and in a culturally relevant way, researchers were directed to examine complex interactions and behaviors simultaneously, and report in a storytelling form as well as in conventional statistics and figures.

Important to the maintenance of relationships with Indigenous partners, it was essential that the research team employed continuous engagement and verification of model iterations and outputs with community representatives at all levels (youth to Elders, Chief and Council members, and community staff). Using agent-based modeling facilitated this, and the project team were able to provide a tool for decision support that had local impacts and furthered knowledge on participatory modeling with Indigenous communities (see Section 5).

Indigenous communities in Canada face higher risks of infections due to environmental and social factors [3,33,41–43]. This work created a context-specific tool that quantified the risks associated with a waterborne disease outbreak through an agent-based model that reflected community dynamics and cultural values. For example, visiting each other within the community is a regular social and cultural practice that promotes wellbeing and connectedness. However, in times of disease outbreak, the social connectedness needs to be offset with public health measures and increased service provision at the health center to protect health and treat illness. Factors such as community dynamics, demographics, and mobility, discoverable with trusting engagement methods, should be carefully considered when developing disease outbreak and climate change mitigation plans. The results of this work demonstrate how agent-based models can draw in these factors through the creation and programming of agents that reflect actual community dynamics. For example, such methods can explicate how people agents moved through the community and/or get blocked by local climate conditions, and when the historically under-supported health stations [9,10] become overwhelmed, resulting in person agents leaving without being seen.

Indigenous and partnered researchers and communities have called for new research tools and approaches that meaningfully include Indigenous peoples' knowledge on health and wellbeing to better conceptualize the health implications of climate change [4,11]. In this model, we co-created a new tool for decision makers in a community that is informed by community-raised concerns for responding to the threats of climate change, specifically flooding and how floods may affect mobility. Our model uniquely combined an accurate representation of the geographic features of the community with detailed flooding information extracted from the WDPM developed by the University of Saskatchewan Centre for Hydrology, and mobility impacts from flooding were further verified by experiences shared by community members. Our model suggests the need to consider flood mitigation measures and flood-proof important roads, create alternate methods of entering/leaving the community for care when needed, and to consider the proportion of community members whose daily activities, which support community wellbeing, may be blocked. Implications include the need for climate change adaptation planning within the community that can be transferred to other threats beyond flooding. This work helps to answer calls by researchers to support Indigenous communities with climate change adaptation research, in appropriate and decolonizing ways [3–7].

Agent-based models developed in collaboration with stakeholders provide a powerful and intuitive tool to explore community-driven concerns, both through the iterative model building process and the model outcomes [22,23]. Community participation in developing the present models of the impacts of water contamination and flooding on the YQFN provided important local understanding and context to the model. For example, community consultation on employment rates for model demographics provided important insight beyond numerical rates. Females experienced greater paid employment rates, with some

males supporting families by unpaid work such as hunting and fishing. People without formal employment also filled additional social and cultural roles, including visiting Elders, clearing new trails, quickly visiting community places to support others, providing cultural programming, or mentoring youth. Representation of this aspect of daily life was important to community members, as the community tends to give the unemployed opportunities to contribute in other ways.

Community input was also important in developing model output. Quantitative output provided understanding for decision makers as to how quickly services would be disrupted or overwhelmed under the different scenarios. The stochastic nature of agent-based models provides further insight through the potential for variability suggesting a range of impacts, importantly allowing decision makers to envision not only average/expected outcomes but more extreme potential outcomes. Implementing the storyteller agent also provided insight on how individual lives might be disrupted. Both forms of output were valuable to the community and more accurately reflect Indigenous worldviews, which tend to be holistic. Further model building and refinement in collaboration with the community is recommended to further explore community-driven concerns around climate change effects, health and wellbeing, and community mobility.

Although the modeling approach taken in this study has many benefits, limitations include the stylized nature of the model, particularly with respect to the waterborne illness that was generalized rather than representing a particular pathogen. In addition, census data were incorporated in the model but are often inaccurate for Indigenous communities due to colonialism (i.e., a history of removal of children and people from the home by settlers means that when unknown individuals knock on doors for census purposes, there is often a refusal to answer), as well as the sometimes transient nature of populations and overcrowding [44,45] Data were drawn from existing published work, and from community records and engagement, however, more data could be incorporated into the model reflecting susceptibility, co-occurring conditions, and frequency of events such as truck contamination, cistern contamination, popular community events, and fluctuations in health center hours and staff availability. In addition, community member population in YQFN surpasses 2400 within the province, however, only around 8–900 people live on the reserve, and the demographics change frequently depending on housing and employment availability. Although developed in close consultation with the community, the parameters informing this model represent a snapshot in time and are not generalizable to other time points or to other communities. In addition, the model was developed before the COVID-19 pandemic and no post-COVID 19 data are available for comparison.

## 5. Conclusions

Indigenous communities and partnering researchers continue to build knowledge on decision support tools that can support resource management and climate adaptation decision making [3–7]. The benefits of this project included incorporating actual needs of the community, using repeated engagement to gather data and co-refine the model, and producing model outputs that aligned with both community cultural traditions (storytelling) and researcher-driven goals. Community-driven questions also drove the model co-design, and members of the community had opportunities to share their ideas and concerns through in-person engagement in their communities when researchers visited. Community research coordinators also developed research-specific skills and assisted the research team in developing interpersonal skills for cross-cultural engagement with Indigenous community members.

In addition, this work has already had local impact; given the predicted blocking of roads due to future flooding, the community elected to build a new lands and maintenance operations building in 2020–2021 with a large, attached storage garage in which a grader, digger, and numerous four-by-four off-road vehicles and snowmobiles are stored at a high, crucial point within the community. Before this work, those vehicles were stored in a town 15 km (about 9.32 mi) away and were inaccessible when needed to transport community

members or obtain supplies. Additionally, new research grants have been secured to develop customized and culturally aligned training programs for water treatment officers working on combined truck-to-cistern and piped distribution systems, not otherwise included in current conventional training programs for water treatment officers. A direct benefit from this context-specific model custom-created for them and with them, was that decision makers reported feeling strengthened in their conviction to make appropriate decisions about climate change adaptations and health service provisions.

Recommendations for communities and community members include seeking out climate predictions and records for their region to inform discussion on climate adaptations. When these data are incorporated into models, such as the agent-based one described here, the combination becomes a powerful decision support tool for directing limited operations and maintenance funding on reserve to solutions for long-term challenges, including infrastructure, flood mitigation measures, and establishing response plans for blocked transport corridors. Communities can and should also have their own local knowledge and traditions included accurately and meaningfully into models so outputs contain details that can inform steps to take for cultural wellbeing and continuity in the face of floods or other climate events. Community leadership can also benefit from these models and the evidence derived from them in negotiating with federal agencies for consistent and increasing support of climate change adaptation measures.

Recommendations for modelers include learning to engage with community members to increase the local relevance and credibility of their work, and to incorporate context-specific and meaningful community dynamics into the models they produce. For example, visiting neighbors is an important and frequent part of daily life and wellbeing in the YQFN community. However, without asking community members directly, this would have gone unnoticed in model parameterization even though it affects disease transmission and community belonging. Considering the types of outputs produced from academic methods is also important; options that provide quantified data are helpful for relations with government agencies, but narrative outputs are also important for cross-cultural communication [46].

Recommendations for researchers include continuing to develop and test modeling software and tools that include customization and context-specific factors for community-partnered work in community health, climate change impacts, and beyond. Committing to multiple iterations of model building with local verification and inclusion of local data will enhance the satisfaction and utility of findings for communities who come forward with community-driven concerns. Researchers can also contribute to joint action with community members by providing data and reports in formats that communities can use when they approach funding agencies, contractors, and others who provide services to communities. In this case, the model outputs were shared with government agents, who agreed that climate and health adaptations were a legitimate investment for community security and wellbeing.

Finally, few other studies that were co-created and based on actual landscapes, climate change, and health threats to Indigenous communities have been published for comparison, or to inform this and future work. At time of submission, no other agent-based models of climate events and health services specific to an Indigenous community in this province, or country could be found. We recommend continued exploration of the utility of co-created agent-based models with Indigenous communities to build both pedagogical frameworks for researcher–community collaborations in the future, and for advancing community efforts for climate adaptation and mitigation.

**Author Contributions:** Conceptualization, L.B. (Lalita Bharadwaj), L.B. (Lori Bradford), M.N., N.D.O. and C.L.W.; community research engagement, participation and relationship building, L.B. (Lalita Bharadwaj), L.B. (Lori Bradford), M.N; data gathering events, L.B. (Lalita Bharadwaj). L.B. (Lori Bradford), M.N, K.B; methodology, L.B. (Lalita Bharadwaj), L.B. (Lori Bradford) and C.L.W.; software, G.W.M.; validation, L.M., G.W.M. and C.L.W.; formal analysis, L.M., G.W.M. and C.L.W.; investigation, K.B., L.B. (Lalita Bharadwaj), L.B. (Lori Bradford), M.N., G.S. and G.W.M.; resources, M.N.; data curation, K.B., L.B. (Lalita Bharadwaj), L.B. (Lori Bradford), and G.S.; writing—original draft preparation, K.B. and G.W.M.; writing—review and editing, L.B. (Lori Bradford), N.D.O., C.L.W., L.M. and G.W.M.; visualization, L.M. and G.W.M.; supervision, L.B. (Lalita Bharadwaj), L.B. (Lori Bradford), N.D.O., G.S. and C.L.W.; project administration, L.B. (Lalita Bharadwaj), L.B. (Lori Bradford), G.S. and C.L.W.; funding acquisition, L.B. (Lalita Bharadwaj), L.B. (Lori Bradford), G.S. and C.L.W. All authors have read and agreed to the published version of the manuscript.

**Funding:** This research was funded by the Canadian First Research Excellence Fund—Global Water Futures under Pillar 1 and titled: PROOF OF CONCEPT Agent Based Modeling as a tool to Investigate Comprehensive Indigenous Health Impacts of Flooding.

**Institutional Review Board Statement:** This study was approved by the Behavioural Research Ethics Committee of the University of Saskatchewan, under certificate 18-08, on 2 July 2018.

**Informed Consent Statement:** Informed consent was obtained from all subjects involved in the study.

**Data Availability Statement:** Data supporting reporting results can be accessed from the Global Water Futures MetaData repository, respecting OCAP® (Ownership, Control, Access, and Possession) at https://gwfnet.net/Metadata/Record/T-2020-11-30-y1DzBd4Z7WEK8529mNxqP1Q accessed date 16 August 2022.

**Acknowledgments:** This work was driven by the Chief and Council, and community members of Yellow Quill First Nation who graciously shared their time, ideas, lived experiences, and concerns with researchers over multiple visits, and pilot tested the model. Technical documents about the community infrastructure, LiDAR data, and demographics were also shared. We are extremely grateful. The authors would further like to thank Jenelle Dreaver who provided culturally relevant icon art for use in the model.

**Conflicts of Interest:** The authors declare no conflict of interest.

## Appendix A

Following is a detailed description of the model, as constructed, following the ODD Protocol [47,48].

### *Appendix A.1. Objectives*

Appendix A.1.1. Purpose

The purpose of the model is to serve as a discussion piece for community engagement by examining possible effects of waterborne illness and flooding on the community of Yellow Quill First Nation. Specifically, impact on the clinic's capacity to serve the community following outbreaks driven by a contaminated water truck and a community event, and impact via transportation limitations of flooding events of varying magnitudes.

Appendix A.1.2. Entities, State Variable, and Scales

The model is made up of People, Places, Water Sources, Water Transporters, Health Conditions, and Ponds. State variables are tabulated in Table A1.

**Table A1.** State variables.

| Name | Description |
| --- | --- |
| Person | |
| Age | The person's age in years |
| Sex | The person's sex |
| Home Location | The Place representing the person's home |
| Work Location | The Place representing the person' workplace, if any |
| Care-seeking probability | Probability the person will seek care if unhealthy |
| Health Conditions | Health Conditions the person has |
| Place | |
| Unique ID | ID of this place |
| Location | Geographic location |
| People Inside | Person agents inside this place |
| Water Source | Water source for the place |
| Water Source | |
| ID | ID of this water source |
| Location | Geographic location |
| Source | Parent water source for the source, if any |
| Reservoir Level | Level of reservoir for the source |
| Secondary Sources | Water sources drawing from the source |
| Water Transporter | |
| Home Base | Water source where the transporter refills |
| Location | Geographic location |
| Reservoir Level | Level of transporter's reservoir |
| Health Condition | |
| Duration | Duration of the condition |
| Symptom Type | Curve shape for symptom severity vs. condition progression |
| Transmission Type | Means of transmission of the condition, if any |
| Transmissivity Begins | Point in progression where transmissivity begins |
| Transmissivity Ends | Point in progression where transmissivity ends |
| Prob of Transmission | Probability of transmission on exposure |
| Full Recovery | Whether the person recover fully at end of condition progression |
| Prob of Death | Probability of death from the condition |
| Curable | Is the condition curable through treatment |
| Treatment Symptom Mult | Multiplier on symptoms due to treatment |
| Treatment Death Multiplier | Multiplier on probability of death due to treatment |
| Immunity Period | Duration of immunity after recovery |
| Treatment Duration | Duration of treatment |
| Pond | |
| Region | The GIS region associated with the pond |
| Collisions | Count of the number of agents that have "collided" with the pond |

Additionally, some aspects of model state are represented by statecharts. Statecharts indicate mutually exclusive states that an agent may occupy, with arrows indicating transitions between these states. Icons on statechart arrows indicate how the transition is triggered: an envelope indicates a message, a flag indicates it is on arrival at a location, a clock denotes a timeout, a question mark symbolizes a condition, and an exponential curve connotes a rate.

A person will move about the community in a manner described by the statechart in Figure A1. A person's movements can include normal commutes, care seeking for health conditions, and attendance at a community cultural event. A person's health is summarized by the health statechart, shown in Figure A2, in which they may be healthy, having no health conditions, or unhealthy, having one or more health conditions. If a person is unhealthy, they may seek care, at a rate of 1.0 per day, according to their care seeking probability.

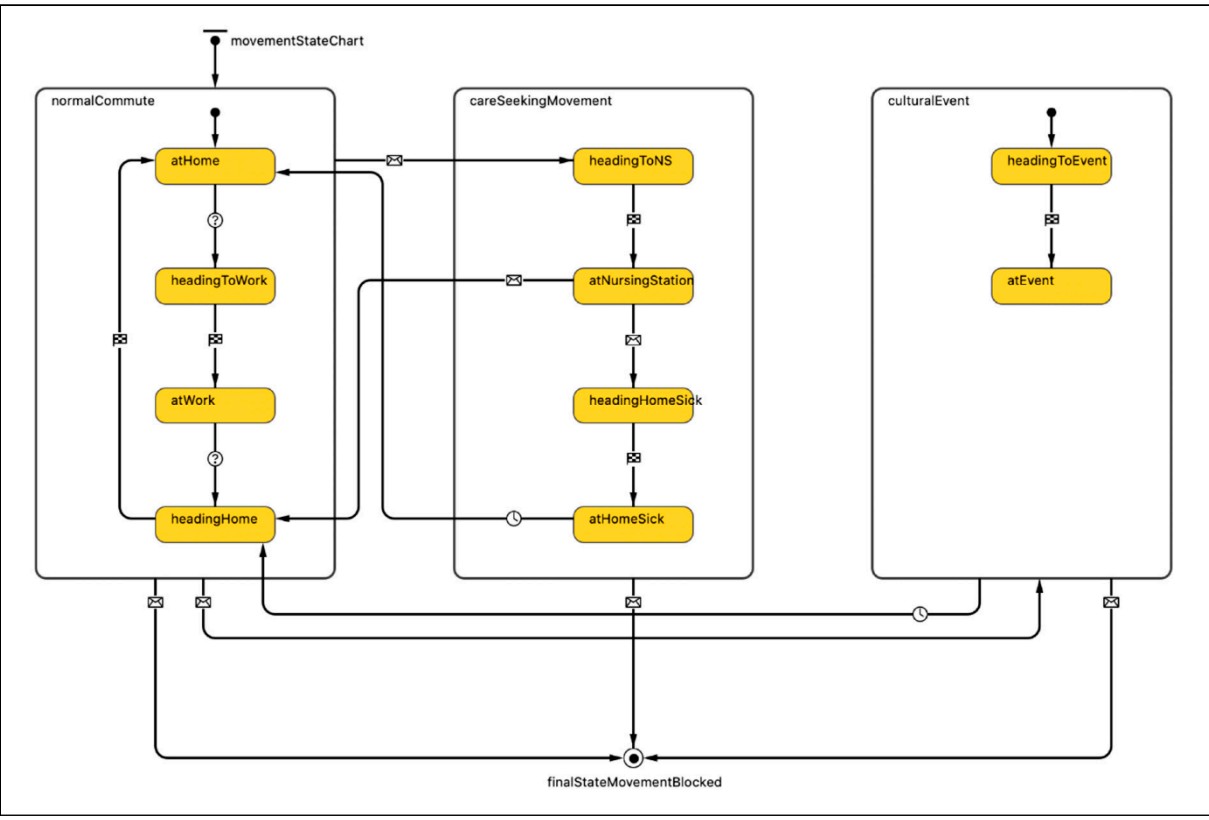

**Figure A1.** Screenshot of movement statechart in the person agent from the AnyLogic editor.

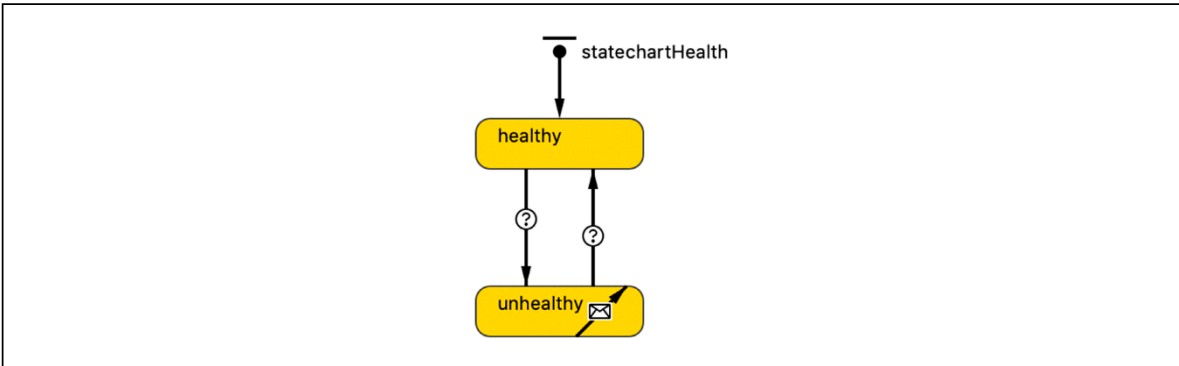

**Figure A2.** Screenshot of the health statechart in the person agent from the AnyLogic editor.

A place agent may be contaminated or clean in terms of pathogens, as represented by the statechart in the bottom left of Figure A3, and may pose any number of generalized health hazards to the occupants; in the current model, this is limited to the risk of general physical injury.

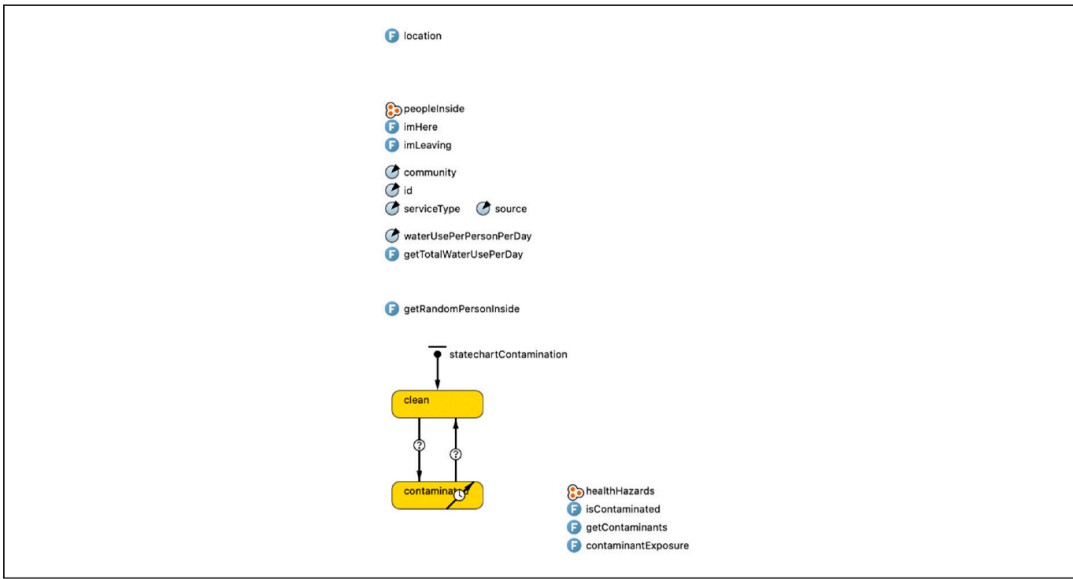

**Figure A3.** Screenshot of the Place agent from the AnyLogic editor, iconography from [29].

In addition to the features common to all places, the nursing station agent features a discrete event simulation (DES) patient flow model, shown in Figure A4. Patients arrive at the enter block on the left and will queue at the allocateBed point until a bed and health worker are available. A person who is waiting may leave without being seen if they wait too long. Once a bed and health worker are assigned to the person, treatment will take a certain amount of time, after which the health worker is free to see other patients. The person may then rest in bed for a time before being released. People who have been released after treatment will self-isolate at home for the duration of their illness but those who leave without being seen will continue about their normal routine, potentially infecting more people.

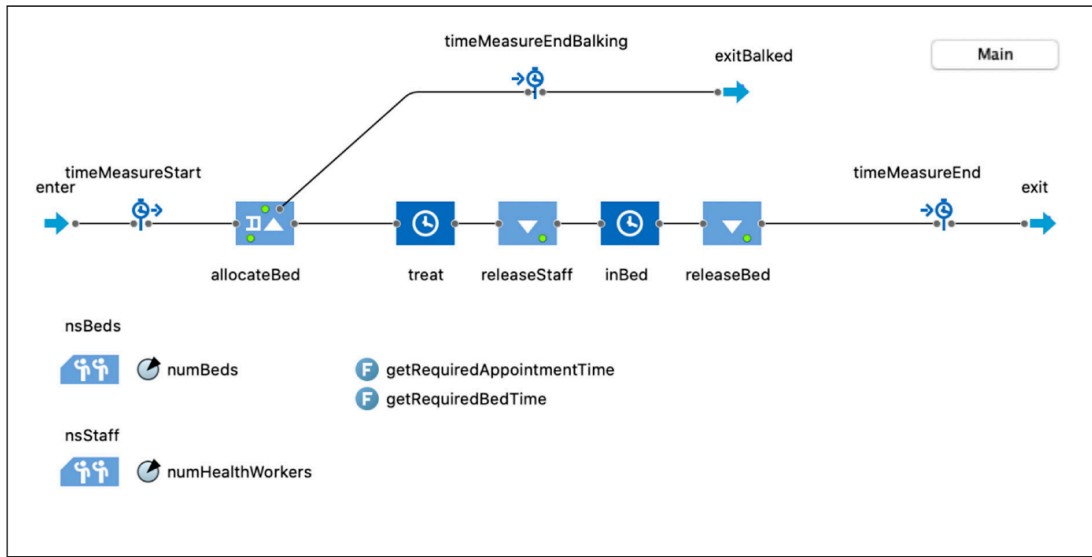

**Figure A4.** Screenshot of patient flow logic in the nursing station agent from the AnyLogic editor, iconography from [29].

The water source agent is general and may represent a water treatment facility, a reservoir, or a cistern serving a single building. Two statecharts govern the state of the water source in addition to the state variables tabulated in Table A1. The calling For Water state applies to cisterns, which will be occasionally filled by delivery truck when their water

level is low. The source may be clean or contaminated with a waterborne pathogen. Sources start out clean and may be contaminated if, for cisterns, contaminated water is delivered. Contamination status does not reset during the timeframe of the model. These statecharts are shown in Figure A5.

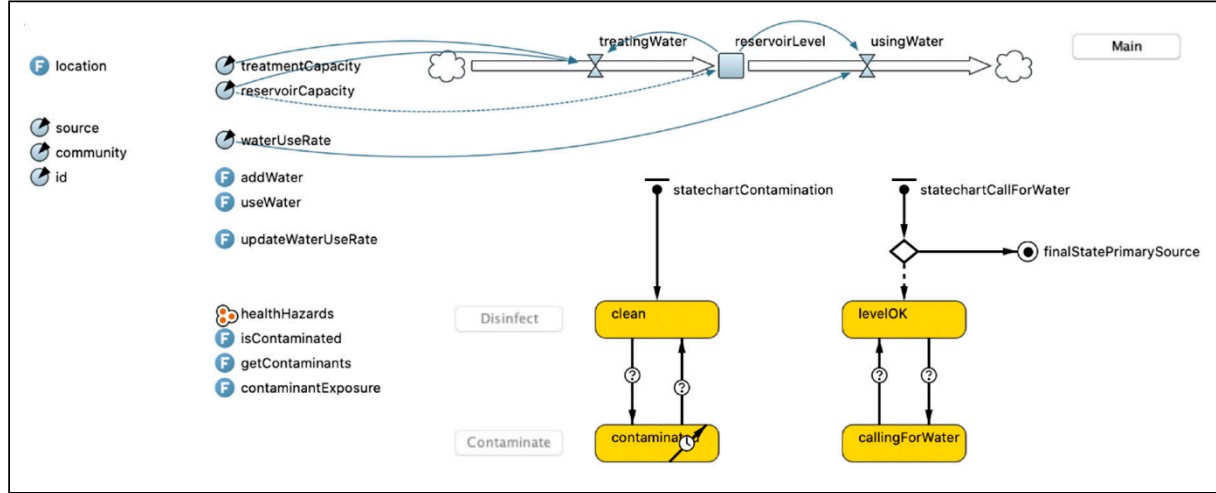

**Figure A5.** Screenshot of the water source agent from the AnyLogic editor, iconography from [29].

The water transport agent represents a water truck that delivers water to houses on a truck-to-cistern system that forms part of the YQFN water distribution system. The water delivery statechart is shown in Figure A6. The transporter will idle when no customers are calling for water and will move to make deliveries in the order the requests are received. The transporter has a specified capacity and will return to the filling station as required. Should the filling station be out of water, the transporter will move to the noWater state and wait for more water to be available.

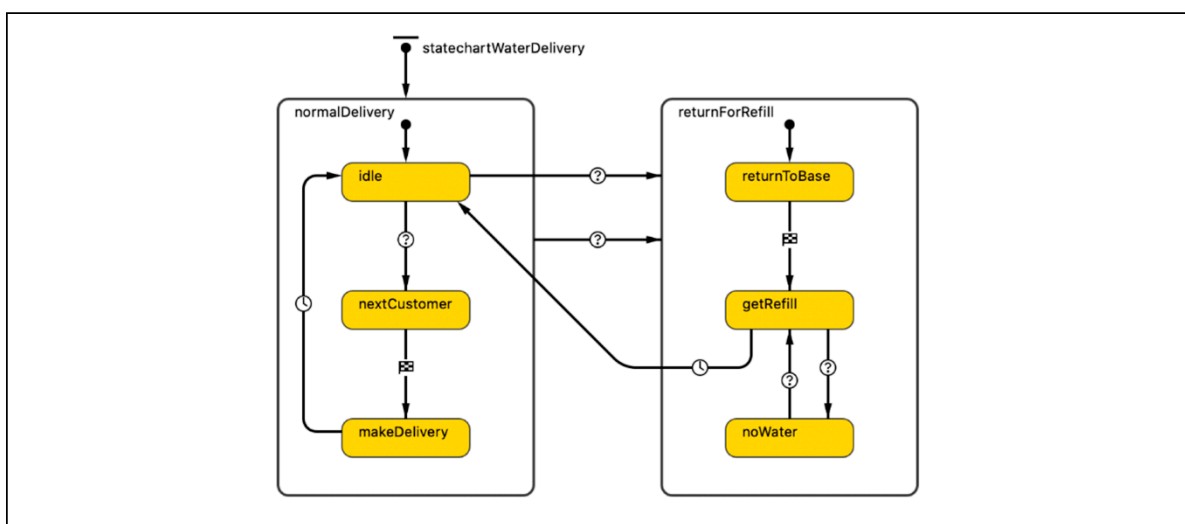

**Figure A6.** Screenshot of water delivery statechart for the water transporter agent from the AnyLogic editor.

A generalized health condition is also represented as an agent. Although it lacks agency, this representation is convenient for allowing editing in the graphical AnyLogic editor and ease of characterizing the dynamics and integration with other agents. The representation is sufficiently generalized to represent many health conditions and is governed by the statecharts shown in Figure A7. For representation of infectious diseases, a health condition may have a pre-transmissible (latent) stage followed by a transmissible stage

then a post-transmissible stage. While transmissible, the person will periodically expose others around them. The health condition will move out of the post-transmissible state after a specified delay or when the person receives the requisite treatment. If a condition is non-communicable, as with injuries and type II diabetes, it will move directly to the post-transmissible state. Chronic diseases will remain in the post-transmissible state permanently. The treatment statechart considers whether the person is currently under the beneficial effects of a treatment, which may optionally reduce symptom severity.

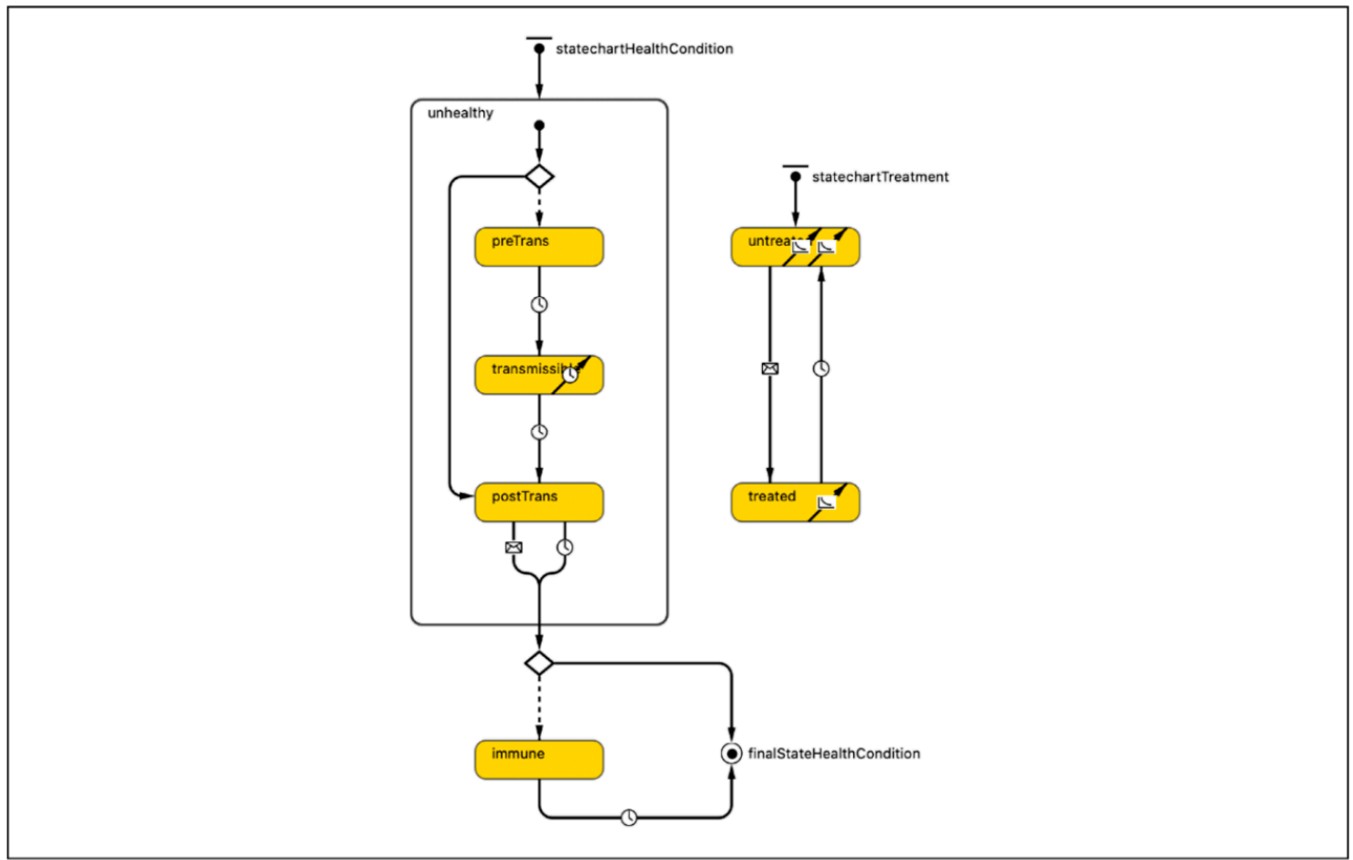

**Figure A7.** Screenshot of the health condition statecharts from the AnyLogic editor.

Appendix A.1.3. Process Overview and Scheduling

AnyLogic provides a continuous-time scheduler where events may occur at any point on the timeline. In the case that events are scheduled at exactly the same time, they are adjudicated in a first-in-first-out order. Time units for this model are days. The contaminated truck and Pow Wow scenarios run for 28 days each and the mobility scenario runs for 5 days.

*Appendix A.2. Design Concepts*

Appendix A.2.1. Basic Principles

This model represents the community of Yellow Quill First Nation, its people, buildings, and water infrastructure. Person agents are mobile and move about the community daily. People may be afflicted with health conditions including a waterborne illness, physical injury, or type II diabetes. Movement of people about the community may be impeded when they encounter ponded water due to flooding.

Appendix A.2.2. Emergence

Key emergent behaviors from this model are the number of people infected with the waterborne illness and the number treated and leaving without being seen at the nursing station in the contaminated truck and Pow Wow scenarios. The number of people whose movement is blocked by flood waters is the key behavior observed in the mobility scenario.

Appendix A.2.3. Adaptation

People adapt to having health conditions by seeking treatment at the nursing station with a certain probability and modifying their routine until recovered. People whose movement is blocked by flood waters stop and declare themselves to be blocked.

Appendix A.2.4. Sensing

The sensing process is not simulated in this model. Agents are assumed to have perfect knowledge of the variables they consider in their decisions.

Appendix A.2.5. Interaction

Water sources may be contaminated if their parent source is contaminated, they are refilled by a contaminated transporter, or as part of the scenario definition. People who spend time in a place with a contaminated water source may contract the waterborne illness. People may also spread the illness to each other when they occupy the same place. People who seek treatment at the nursing station may leave without being seen if the wait times are too long; wait times are determined by the number of people seeking care. People who attempt to cross ponded flood waters will have their movement blocked.

Appendix A.2.6. Stochasticity

The model is initialized stochastically to allow for variability between runs. All rate transitions in statecharts are computed stochastically with the rate representing the probability per unit time of making the transition. Transmission of communicable diseases is stochastic following a probability of transmission. AnyLogic allows for repeatable runs by setting a fixed seed for the common pseudorandom number generator.

Appendix A.2.7. Collectives

Places in this model can be considered collectives. People inside a place may transmit communicable disease to one another. If the place has a contaminated water source, the people inside may contract a waterborne illness. The number of people inside a place determine its rate of water consumption.

Appendix A.2.8. Observation

The model outputs, graphically, plots of several measures over time from the simulation, shown in Figure A8. One presents the fractions of the population that are susceptible, exposed, infectious, and recovered with respect to a waterborne illness. Current counts of people with each health status are displayed as well as cumulative counts of people treated at the nursing station for each status. Cumulative counts of people treated and people who leave without being seen (LWBS) at the nursing station are also displayed.

For the mobility scenario, the model also displays the fraction of people in the general community whose movement is blocked by flooding and the count of people unable to seek care for type II diabetes, shown in Figure A9.

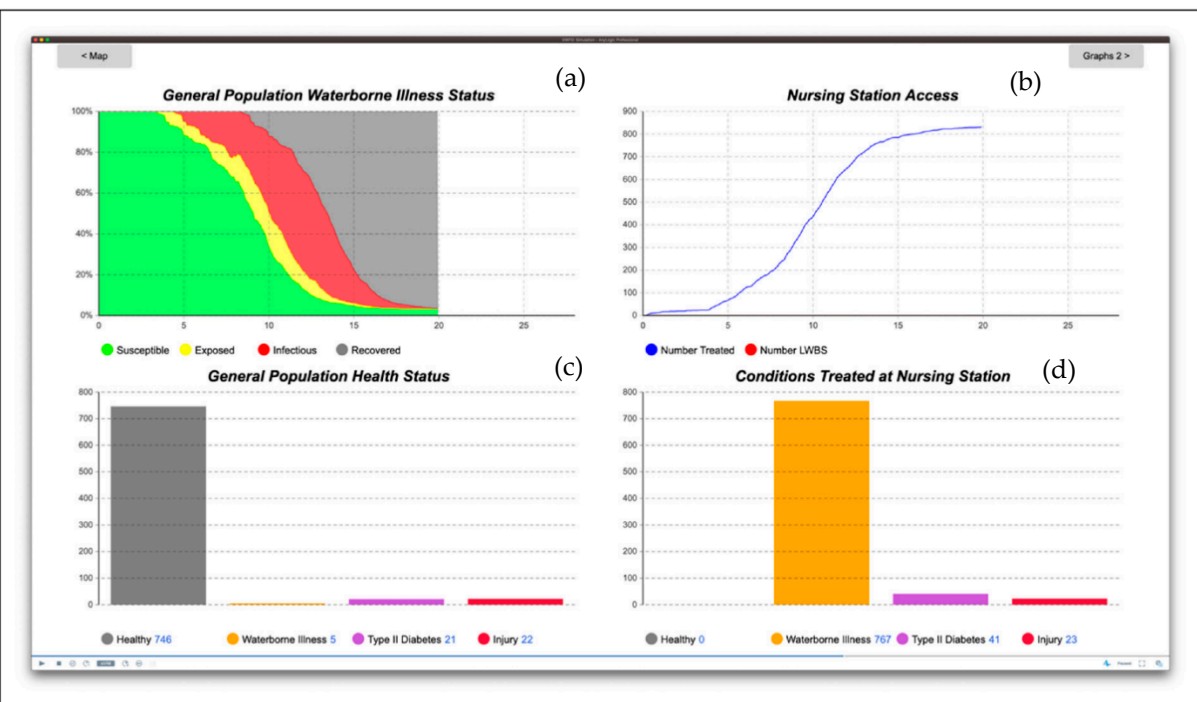

**Figure A8.** Screenshot of dynamic model outputs including (**a**) the status of waterborne illness in the population (proportion susceptible, exposed, infectious, recovered), (**b**) counts of the population experiencing each health condition, (**c**) cumulative counts of people treated or LWBS at the nursing station, and (**d**) counts of people seen with each health condition at the nursing station.

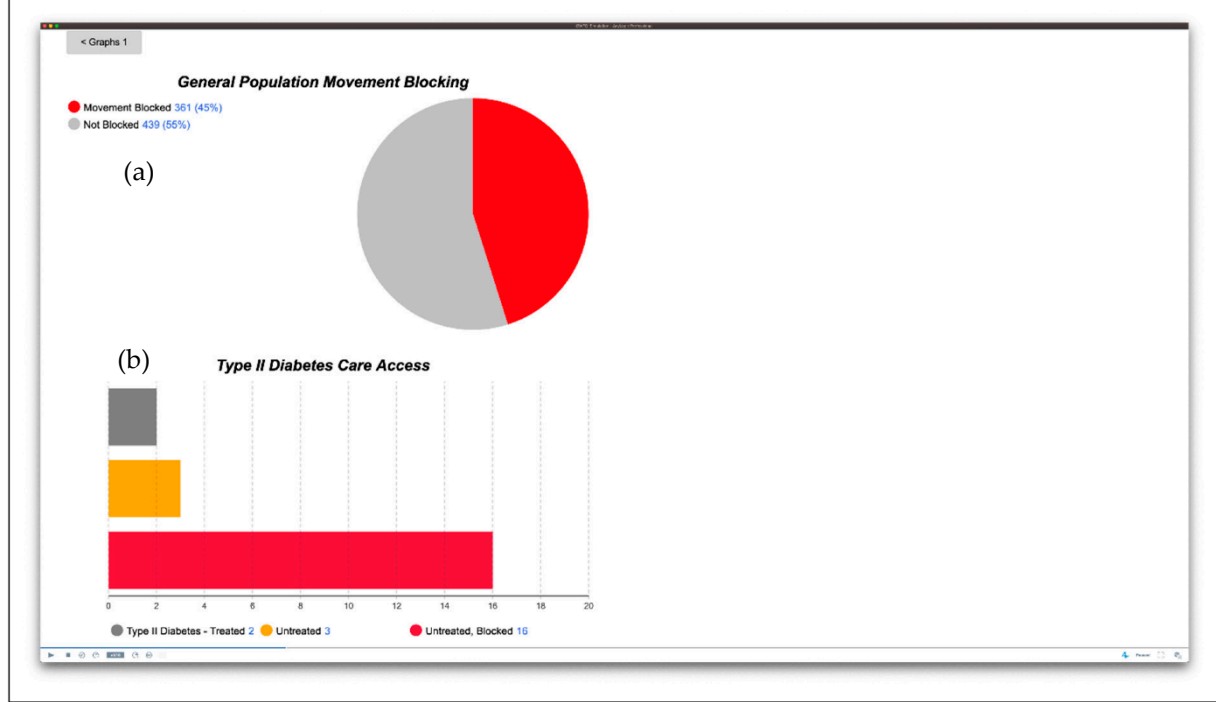

**Figure A9.** Screenshot of dynamic model outputs for mobility scenario including the proportion of population whose movement is blocked (**a**) and counts of persons with diabetes treated, untreated, and untreated due to movement being blocked (**b**).

For Monte Carlo ensembles, the values of these outputs at the end of the simulation for each individual realization is exported in Excel format.

*Appendix A.3. Details*

Appendix A.3.1. Initialization

Initialization is predominantly generic and takes place in the following order. Places and water infrastructure are initialized, loading locations and other properties from files. Health conditions are loaded from a file. Flooded areas are initialized by loading the GIS polygons from a file and instantiating pond agents. The population is initialized stochastically by drawing the person's sex, then drawing their age from a sex-specific age distribution, then assigning a home at random. If the person is employed in paid work, they are also assigned a workplace at random. The water network is then set up by connecting places and water sources. Storytellers are then set up and associated with person agents at random. Finally, the run time is set to a scenario-specific value and the event that triggers the start of the scenario is scheduled.

Appendix A.3.2. Input Data

The model does not use input data to represent time-varying processes.

Appendix A.3.3. Parameters

Parameters are specified using parameter objects within the AnyLogic editor. For example, floodLevel indicates which flooding level to use to mark ponded areas, fracDiabetics indicates the prevalence of type II diabetics requiring dialysis within the population, numWaterTrucks indicates the number of water trucks serving the community, and populationSize is the number of people in the community. The scenario parameter indicates which scenario is to be run. The ages of agents are drawn from a probability distribution derived from population pyramid from 2016 (year) Census of Population, Canada [10] and sexRatio is the ratio of the number of male to female people in the community. Some additional parameters are used for user interface control but do not influence the simulation outcomes. Parameters are listed in Tables 2, 3 and A2.

Additionally, some setup data are loaded from files. Locations and properties of Home, Workplace, NursingStation, School, WaterSource, and CulturalSite agents are loaded from a spreadsheet. The spreadsheet was generated using QGIS software [49] by locating the structures on an air photo of YQFN and adding metadata provided by the community. Ponded areas due to flooding are loaded from a GeoJSON file, as discussed in Appendix A.2. HealthConditions are loaded from a file as well, to allow for specification of new conditions without changing programming.

**Table A2.** Additional Model Parameters.

| Parameter | Default Value |
| --- | --- |
| numWaterTrucks | 1 |
| probability cistern contaminated by truck | 1 |
| waterUsePerPersonPerDay (cistern) | 200 L/person/day [5] |
| waterUsePerPersonPerDay (piped) | 500 L/person/day |
| fracDiabetics | 0.02 |
| isShowingBuildings | TRUE |
| isShowingWaterSources | TRUE |
| isShowingPeople | TRUE |
| scenario | FREEFORM |
| isUsingAltArt | FALSE |
| isShowingLegend | TRUE |
| iconScaleFactor | 1.0 |
| slowForAgentMovements | FALSE |
| slowTimeScale | 0.002 |
| fastTimeScale | 0.1 |

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
