# Peer review of "Case Study of Collaborative Modeling in an Indigenous Community"

_water, doi:10.3390/w14172601_

Round 1

Reviewer 1 Report

The paper describes a collaborative modeling project in the context of an Indigenous community.

The model aims to explore various scenarios related to water security and health.

The main point of the article is related to the model's purpose.

As expressed in Section 2.3.1, it is mainly related to water security and health, through the 

study of three scenarios: (i) effects  of microbial contamination of water delivery truck; 

(ii) spread of waterborne pathogens throughout the community if contaminated water were to be served at a Pow Wow; (iii) impact of flooding on preventing access to health care in the community due to ponding and road washouts.

Whereas Scenario (i) and (ii) require epidemiological spread models, scenario (iii) requires a hazard and crisis-management model.

As a consequence, these 3 scenarios are not really 3 scenarios on the same model, but rather at least  2 different models (or even 3 models).

What they have in common is the health care center and its limited access, which means that, if  the paper claims to present 1 single model, the key purpose should be focused on that point.

Concerning the presentation of the model, the authors choose to not rely on any description standard or protocol (such as the O.D.D. protocol, Volker et al.).

As a consequence, Section 2.3.2 mixes the presentation of the definition of agents, and the dynamics of the model; which is also presented in the 2 following sections. In addition, some sections presenting input data or the stochasticity of the model are missing.

The presentation of the model needs thus to be improved. The use of a protocol could be a convenient way to do it.

The third point is the presentation of the implication of the community in the work.

In every section of the paper, the authors acknowledge the key role and impact of the community in the verification of models, and data ... This is a very interesting point of the paper, but the way the community is involved in the modeling process is not documented and analyzed carefully.

As an example, on page 8. It is stated: "Agents in the model ... verified by community members."

Verification of models is always a  very important point for ABM. 

It would be interesting to have a clear description of how it has been verified, what is the workflow ...

More generally, on the one hand, the text is made heavy with mentions in many places of the involvement of the community, and on the other hand, the involvement method is not made precise enough.

It would be more interesting to dedicate one or two sections to describing carefully and formalizing the way the community was involved in the process. The methods that have been applied could be mentioned and a discussion about their application could be interesting. Answers to questions such as: How are the models verified by the community? How data are gathered from the community? What is the status of  these data related to scientific or statistical data ? how the quality of these data are evaluated ? ...

From the modeling task point of view, it seems  that modelers have conducted a co-modeling process and tried to integrate in one model, all the concerns elicited  from workshops. This produces one single model claiming to simulate several scenarios, that seems to be  more 2 or 3 different models.

It  would have  been  more  interesting to choose between the various problems elicited with the community,  and  build a model  to focus on a single question.  

The context section is very short and does not provide all the necessary information needed to well-understand the elements introduced in the models.

Pages 3 & 4: definition of ABM:

* "unique, in that they have one or  more characteristics that differ from agent-to-agent": several agents can have the same values for each characteristic. But ABM introduces this possibility to have heterogeneity among agents.

* "at a logical or physical position" : social would be better that logical

* "act independently from one other" : agents do not act independently from one other. They can make their own decisions, influenced by their perception of the physical and  social environment (and thus influenced by other agents), their own objectives...

Page 5. Section 2.3. GIS data is used to initialize the simulation. The use of precise GIS, in a simplified model used for discussion, should be discussed.

Page 5. "discrete event simulation model": ABM is introduced; Discrete-event model should also be.

Page 7. A "story telling" module is presented. But there is not a precise description on how it is used in the experiments with the community, or an analysis of what it brings.

Page 8. Section 2.3.5. 

Why has type II diabetes been chosen as a disease to be modeled ? Is it in any case related to water?

Or the number of cases is particularly high in this community. If it  is the case, it should be introduced in the Context section.

Page 1. The section 1.1 (starting the paper) acknowledges  privileges and introduces  the involvement of the community in the research. The section should be better in section 2, in the material and methods and  extended to better describe the social science ground of  this position and how it will influence the scientific work. Similarly section 2.1 should either be moved to the end  of  the paper (close  to acknowledgment or in the metadata of the paper, depending on the journal.)

Author Response

Re: Peer Review of Case Study of Collaborative Modeling in an Indigenous Community

Dear Editor and Reviewers,

We thank you for taking the time to review our manuscript. As requested, we will address each of the reviewers’ concerns point-by-point below.

We look forward to moving this publication forward and we are grateful for the comments of the reviewers which have substantially improved this piece.

Sincerely,

Lori Bradford and Wade McDonald on behalf of the research team.

Reviewer 1:

Concern:

“…As a consequence, these 3 scenarios are not really 3 scenarios on the same model, but rather at least 2 different models (or even 3 models). What they have in common is the health care center and its limited access, which means that, if the paper claims to present 1 single model, the key purpose should be focused on that point.”

Response:

We believe we are using the term ‘model’ in a more operational sense than the reviewer. It is true that the contaminated truck and Pow Wow scenarios include a representation of epidemiology that is ignored in the mobility scenario which, likewise, includes a representation of flood water ponding is ignored in the former two scenarios. The simulations of these scenarios share much more in common, however, including the representation of people, buildings, and infrastructure in the community. Including both representations in the same simulation allows for possible future examination of combined scenarios. We have added a statement in Section 2.4.1 to emphasize the central role of health care access in this work.

Concern:

“Concerning the presentation of the model, the authors choose to not rely on any description standard or protocol (such as the O.D.D. protocol, Volker et al.)…”

Response:

Appendix A now follows the ODD Protocol.

Concern:

“…It would be more interesting to dedicate one or two sections to describing carefully and formalizing the way the community was involved in the process…”

Response:

The description of community engagement has been enhanced in lines 138-167. This particular manuscript is about the model development as a tool. Future manuscripts will explore the process of creating the model together as a form of transdisciplinarity, and more specific analyses of the particular data (asset maps, interviews, etc.). 

Concern:

“The context section is very short and does not provide all the necessary information needed to well-understand the elements introduced in the models.”

Response:

The context of Yellow Quill First Nation is described in Section 1.2 lines 42-57, and includes site characteristics, community concerns, the academic need for this collaborative work, and some of the policy concerns (chronic underfunding, preparation for climate change, creation of culturally-informed tools). Section 2,2 has also been expanded to discuss the characteristics of community engagement that occurred. Could the review please specify addition contextualizing factors needed, and we would be happy to comply.

Concern:

“Pages 3 & 4: definition of ABM”

Response

Wording has been changed in Sec. 1.3.

Concern:

“The use of precise GIS, in a simplified model used for discussion, should be discussed.”

Response:

We have added a paragraph to Section 2.3.3 discussing this.

Concern:

“ABM is introduced; Discrete-event model should also be.”

Response:

Section 1.4 has been added, describing discrete-event simulation.

Concern:

“A ‘story telling’ module is presented. But there is not a precise description on how it is used in the experiments with the community, or an analysis of what it brings.”

Response:

This has been explained in additional detail in Section 2.3.2.

Concern:

“Why has type II diabetes been chosen as a disease to be modeled?”

Response:

Type II diabetes disproportionately impacts Indigenous peoples in Canada and was of interest to the community. Text has been added to Section 2.3.5 to emphasize this.

Concern:

“The section 1.1 (starting the paper) acknowledges privileges and introduces the involvement of the community in the research. The section should be better in section 2, in the material and methods and extended to better describe the social science ground of this position and how it will influence the scientific work.”

Response:

We appreciate that the reviewer noticed our Treaty acknowledgment, positionality statements and recognition of privilege. We believe the location of the Treaty acknowledgement, and positionality early on – in Section 1.1 - is necessary because it serves to identify how our team is working to answer the calls to all Canadians to improve relations with Indigenous communities through the Truth and Reconciliation Commission of Canada. We do this by noting that Indigenous Peoples were here first on these lands and came before any of our western scientific advancements did, and hence, should be given primacy in our scientific manuscripts, especially in research partnered with Indigenous communities. We also do this to identify potential bias, alert the reader to our relations with partners as being equal and reflective, and reduce potential marginalization of partners in this work as subjects of the work instead of partners. For more about the importance of this in the context of Canadian research, please see

https://onlinelibrary.wiley.com/doi/full/10.1002/jee.20377

Concern:

“Similarly section 2.1 should either be moved to the end of the paper (close to acknowledgment or in the metadata of the paper, depending on the journal.)”

Response:

We believe that Sections 1.1 and 2.1 are positioned appropriately and invite the editor to comment.

Reviewer 2:

Concern:

“The advantages and disadvantages of the different methods can be compared in the form of a table, so that the reader will have a clearer understanding of the reasons for using the ABM method.”

Response:

Table 1.1 has been added.

Concern:

According to the decision-making behavior, ABM model can be divided into two categories: reactive and thinking, of which the thinking agent has a stronger ability to portray the behavior of the subject, while the reactive type has higher flexibility and fault tolerance. Is it necessary to clarify?

Response:

This model can be considered reactive. A sentence has been added to Section 2.3 to address this question.

Concern:

“…Some literature points out that in the process of constructing the agent interaction mechanism, the environmental network includes geographical location distance, and the social distance caused by economic, cultural, and ethnic differences, is it necessary to expand the factors that distinguish agents?”

Response:

Geographic location is considered in this model. Economic, cultural, and ethnic differences were not considered but remain open as possibilities for future work should community interest and data availability permit. The wording in Section 2.3.2 has been revised.

Concern:

“Water management ABM often does not have effective predictive capabilities, can not provide practical information for decision-makers, this article also focuses on simulating the impact of water pollution in different scenarios on the community, the future needs to further improve the ABM parameter calibration, result verification, so that the article can better serve the practice of water resources management.”

Response:

Text and references addressing this concern were added to section 2.3.1.

Concern:

“…the lines of some pictures are white, the recognition is not very high, you can consider replacing a more eye-catching color…”

Response:

Figures have been reformatted.

Reviewer 3:

Concern:

“The representation / formatting of the figures and graphics in the main text in the pdf was not always readable…”

Response:

Figures have been reformatted.

Concern:

“One area I was less clear about and how it 'worked' in practice is the storytelling element in the model.”

Response:

The storytelling module has been explained in additional detail in Section 2.3.2.

Reviewer 2 Report

This manuscript used a community engaged approach to co-create an agent-based model geographically bounded to a reserve community to examine three community-requested simulations,which gives recommendations for communities, researchers, and modelers.A lot of work has been carried in this article,but there are some shortcomings,the specific comments are as follows:

Comment 1:The advantages and disadvantages of the different methods can be compared in the form of a table, so that the reader will have a clearer understanding of the reasons for using the ABM method.

Comment 2:According to the decision-making behavior, ABM model can be divided into two categories: reactive and thinking, of which the thinking agent has a stronger ability to portray the behavior of the subject, while the reactive type has higher flexibility and fault tolerance. Is it necessary to clarify?

Comment 3:In the agent section this article is distinguished by age, gender, family, and place of work. Some literature points out that in the process of constructing the agent interaction mechanism, the environmental network includes geographical location distance, and the social distance caused by economic, cultural, and ethnic differences, is it necessary to expand the factors that distinguish agents?

Comment 4:Water management ABM often does not have effective predictive capabilities, can not provide practical information for decision-makers, this article also focuses on simulating the impact of water pollution in different scenarios on the community, the future needs to further improve the ABM parameter calibration, result verification, so that the article can better serve the practice of water resources management.You can refer to the following article:

[1]  A framework for an agent-based model to manage water resources conflicts[J]. Water resources management, 2013, 27(11): 4039-4052.

Comment 5:For the running out of the picture needs to be a clearer explanation, as shown in Figure 3.2, the lines of some pictures are white, the recognition is not very high, you can consider replacing a more eye-catching color, you can refer to the following article:

[2]  Trade-offs in land-use competition and sustainable land development in the North China Plain[J]. Technological Forecasting and Social Change, 2019

Author Response

(The authors gave the same response as above.)

Reviewer 3 Report

This is an well written paper on a very relevant and important topic.  The authors report on well designed, co-researched and co-modelled agent-based scenarios on several aspects of water governance in an indigenous community.  The conceptualisation and implementation of the research is very sensitive to the community and has clearly had significant impacts.  Of particular note in this paper is the authors' commitment to the modelling as a learning process for themselves and the community  - a commitment which has built and maintained trust between the researchers and the community (though recognising this boundary description is not always accurate). 

The representation / formatting of the figures and graphics in the main text in the pdf was not always readable: some figures were 'half off the page' and unreadable.  For some of the graphics, the curves where shown in white which meant they/keys were very difficult to read.  However, I am happy to accept that the graphics will be accurate reflection of the findings. 

One area I was less clear about and how it 'worked' in practice is the storytelling element in the model.  The authors could review the paper and see if this aspect could be explained and demonstrated more clearly, especially in the 'results' and discussion thereof.  I did not see much sense of storytelling in the paper (other than the implicit narrative explaining how different parts linked - was this the storytelling?).  This is not a critical requirement, but a brief, more explicit commentary could help readers understand how this was done and how it can be reported.  

I look forward to seeing the final paper in print.

(Apologies for the delay in the review - I had Covid recently and am only just catching up)

Author Response

(The authors gave the same response as above.)
